# Exact Recovery Guarantees for Parameterized Nonlinear System Identification Problem under Sparse Disturbances or Semi-Oblivious Attacks

**Haixiang Zhang** *haixiang_zhang@berkeley.edu*
*Department of Mathematics*
*University of California, Berkeley*

**Baturalp Yalcin** *baturalp_yalcin@berkeley.edu*
*Department of IEOR*
*University of California, Berkeley*

**Javad Lavaei** *lavaei@berkeley.edu*
*Department of IEOR*
*University of California, Berkeley*

**Eduardo Sontag** *sontag@gmail.com*
*Department of ECE and BioE*
*Northeastern University*

**Reviewed on OpenReview:** *https://openreview.net/forum?id=c9o9UAmN3r*

## Abstract

In this work, we study the problem of learning a nonlinear dynamical system by parameterizing its dynamics using basis functions. We assume that disturbances occur at each time step with an arbitrary probability $p$, which models the sparsity level of the disturbance vectors over time. These disturbances are drawn from an arbitrary, unknown probability distribution, which may depend on past disturbances, provided that it satisfies a zero-mean assumption. The primary objective of this paper is to learn the system's dynamics within a finite time and analyze the sample complexity as a function of $p$. To achieve this, we examine a LASSO-type non-smooth estimator and establish necessary and sufficient conditions for its well-specifiedness and the uniqueness of the global solution to the underlying optimization problem. We then provide exact recovery guarantees for the estimator under two distinct conditions: boundedness and Lipschitz continuity of the basis functions. We show that finite-time exact recovery is achieved with high probability, even when $p$ approaches 1. Unlike prior works, which primarily focus on independent and identically distributed (i.i.d.) disturbances and provide only asymptotic guarantees for system learning, this study presents the first finite-time analysis of nonlinear dynamical systems under a highly general disturbance model. Our framework allows for possible temporal correlations in the disturbances and accommodates semi-oblivious adversarial attacks, significantly broadening the scope of existing theoretical results.

## 1 Introduction

Dynamical systems serve as the foundation for several fields, including sequential decision-making, reinforcement learning, control theory, and recurrent neural networks. They are essential for analyzing and controlling the behavior of real-world physical systems. However, accurately modeling dynamical systems is challenging due to their complexity and the large-scale nature of modern systems. The problem of estimating or learning a system's dynamics from past observations is known as the *system identification* problem.

This problem is extensively studied in the control theory literature, typically under the restrictive assumption that disturbances are small in magnitude and follow an independent and identically distributed (i.i.d.) probability distribution, accounting for modeling errors, measurement noise, and sensor inaccuracies. In contrast to the conventional small-value, dense-over-time i.i.d. disturbance model, this paper considers a sparse-over-time, large-value, and temporally correlated disturbance model. Specifically, we analyze settings where the disturbance is often zero, but when nonzero, it can take large values and exhibit correlations with past disturbances.

The primary motivation for this work arises from emerging safety-critical applications, such as smart grids, autonomous vehicles, and unmanned aerial vehicles, which require robust estimation of system dynamics in the presence of sparse but large and potentially adversarial disturbances. While machine learning techniques have demonstrated significant success in various domains, such as computer vision and natural language processing, their application to safety-critical systems remains limited due to a lack of theoretical guarantees. This paper addresses this gap by providing strong theoretical results for learning dynamical systems using machine learning techniques.

As a motivating example, we consider the dynamical system associated with a power grid, such as the U.S. electrical grid or a regional interconnection. The system states capture various physical parameters, including voltage magnitudes and frequencies across different parts of the network. To enhance the sustainability, resiliency, and efficiency of energy systems, modern power grids integrate large volumes of renewable energy sources, such as wind turbines, solar panels, and electric vehicles. The operation of power systems has become increasingly complex due to the active participation of consumers, who strategically respond to electricity prices by adjusting their consumption based on price signals. Simultaneously, the widespread deployment of sensors across the grid has enabled data-driven grid operation by continuously collecting and analyzing system data. However, this advancement introduces a significant vulnerability: even a tiny, strategically executed data manipulation could mislead power suppliers, causing them to overestimate or underestimate electricity demand. Such miscalculations could result in severe consequences, including system-wide blackouts. This scenario can be modeled as a nonlinear dynamical system in which the system input is subject to *semi-oblivious attacks* at various locations, resulting in the injection of incorrect electricity values into the grid. Given the integration of a large number of new devices into the system, coupled with strategic human behavior, power operators lack a complete model of the underlying dynamical system. Consequently, they must simultaneously learn the system dynamics and detect potential adversarial attacks to mitigate disruptions and restore regular operations effectively. If an input attack remains unaddressed, it can destabilize the system's transient behavior, leading to signal instability and potentially triggering a cascading failure across the grid.

Prior research on robust system learning has primarily focused on unreliable measurements, where the objective is to extract knowledge from noisy and corrupted observations—such as in the matrix sensing problem in machine learning. However, this paper addresses an emerging and largely overlooked aspect of robust learning in safety-critical systems, where the system input itself is manipulated (potentially by an adversary), thereby affecting the system states and inducing instability. Existing results in system identification have concentrated mainly on the asymptotic properties of the least squares estimator (LSE) (Chen & Guo, 2012; Ljung et al., 1999; Ljung & Wahlberg, 1992; Bauer et al., 1999). With the advent of statistical learning theory, research in this area has evolved to study the required number of samples necessary to achieve a given error threshold (Tsiamis et al., 2023). While early non-asymptotic analyses focused on linear time-invariant (LTI) systems with i.i.d. disturbances using mixing arguments (Kuznetsov & Mohri, 2017; Rostamizadeh & Mohri, 2007), more recent studies employ martingale and small-ball techniques to derive sample complexity guarantees for LTI systems (Simchowitz et al., 2018; Faradonbeh et al., 2018; Tsiamis & Pappas, 2019). For nonlinear systems, parameterized models have been explored in recent studies (Noël & Kerschen, 2017; Nowak, 2002; Foster et al., 2020; Sattar & Oymak, 2022; Ziemann et al., 2022), demonstrating the convergence of recursive and gradient-based algorithms to the true parameters with a convergence rate of $T^{-1/2}$ using martingale techniques and mixing time arguments. These results mainly focused on the non-asymptotic *approximate* recovery of the system. On the other hand, we are more interested in the non-asymptotic *exact* recovery guarantees. As early attempts toward this goal, there has been some progress toward developing non-smooth estimators for both linear and nonlinear systems (Feng & Lavaei, 2021; Feng et al., 2023; Yalcin

et al., 2023), particularly in handling large-but-sparse noise vectors with dependencies. However, robust regression techniques incorporating regularization (Xu et al., 2009; Bertsimas & Copenhaver, 2018; Huang et al., 2016) remain relatively unexplored in the context of dynamical systems, particularly with respect to non-asymptotic sample complexity analysis. This gap is mainly due to the inherent autocorrelation in system samples, making traditional statistical tools less applicable. A more detailed literature review is provided in Section 2.

This paper lays the foundation for advancing online optimal control in the presence of large but sparse disturbances, such as specific adversarial attacks. Under safety-critical constraints, it is generally necessary to first learn the system dynamics before applying a control strategy, since applying an inadequate policy could shift the system states beyond safe limits, potentially leading to instability; see Section 2 and Moerland et al. (2023) for more detailed discussions. Therefore, to avoid the potential risks, a crucial first step in achieving the goal of optimal control under adversarial attacks is accurately learning the system dynamics. To this end, we focus on the system identification problem for parameterized nonlinear systems, making necessary assumptions about the disturbance model, which will be discussed in later sections. We model the unknown nonlinear functions describing the system via a linear combination of some given basis functions by taking advantage of their representation properties. Our objective is to estimate the parameters of these basis functions, which dictate the updates of the dynamical system. Formally, we consider the following autonomous dynamical system:

$$x_0 = \mathbf{0}_n, \quad x_{t+1} = \bar{A}f(x_t) + \bar{d}_t, \quad \forall t \in \{0, \dots, T-1\}, \tag{1}$$

where $f : \mathbb{R}^n \mapsto \mathbb{R}^m$ is a combination of $m$ known basis functions and $\bar{A} \in \mathbb{R}^{n \times m}$ is the unknown matrix of system parameters. The system trajectory is also influenced by the disturbance term $\bar{d}_t \in \mathbb{R}^n$. We consider a *sparse-but-large* disturbance scenario, where the disturbance is sometimes zero but, when nonzero, takes large values. In the context of an adversarial attack, a zero disturbance value for $\bar{d}_t$ indicates the absence of attack at time $t$, while a nonzero disturbance is typically large, designed to maximize the impact of the attack. The goal of the system identification problem is to recover the ground truth matrix $\bar{A}$ using observations of the system states, i.e., $\{x_0, \dots, x_T\}$. The model generating $\bar{d}_t$'s is unknown to the user; however, to ensure unique recoverability of the system from measurements, we will later introduce necessary assumptions on the disturbance model, referred to as *semi-oblivious attacks*.

Unlike empirical risk minimization problems, where samples are typically assumed to be i.i.d., the system states $\{x_0, \dots, x_T\}$ exhibit auto-correlation. As a result, the standard i.i.d. assumption on the data-generating distribution is violated. This auto-correlation introduces significant theoretical challenges, which we address in this work by proposing a novel and nontrivial extension of exact recovery guarantees to the system identification problem. Since the disturbance $\bar{d}_t$ is unknown to the system operator and can take a large value, it is necessary to utilize estimators to the ground truth $\bar{A}$ that are robust to variations in $\bar{d}_t$ and converge to $\bar{A}$ within a finite time horizon $T$. Our work is inspired by Yalcin et al. (2023) that studied the above problem for linear systems. The linear case is noticeably more straightforward than the nonlinear system identification problem since each observation $x_t$ becomes a linear function of previous disturbances. In contrast, for nonlinear systems, the relationship between measurements and disturbances is considerably more complex, requiring substantial technical advancements beyond the methods used in Yalcin et al. (2023).

The existing literature has primarily focused on the smooth least-squares estimator:

$$\hat{A} \in \arg \min_{A \in \mathbb{R}^{n \times m}} \sum_{t=0}^{T-1} \|x_{t+1} - Af(x_t)\|_2^2. \tag{2}$$

In contrast, motivated by the exact recovery properties of non-smooth loss functions (e.g., the $\ell_1$-norm and the nuclear norm), we consider the following alternative estimator:

$$\hat{A} \in \arg \min_{A \in \mathbb{R}^{n \times m}} \sum_{t=0}^{T-1} \|x_{t+1} - Af(x_t)\|_2. \tag{3}$$

We note that the optimization problem (3) remains convex in $A$ despite its non-smooth objective function; therefore, it can be solved efficiently by existing optimization solvers. The estimator (3) is closely related

to the LASSO estimator, as its loss function can be interpreted as a generalization of the $\ell_1$-loss function. More specifically, when $n = 1$, the estimator (3) simplifies to

$$\hat{A} \in \arg \min_{A \in \mathbb{R}^{1 \times m}} \sum_{t=0}^{T-1} |x_{t+1} - Af(x_t)|,$$

which corresponds to the auto-correlated general linear regression estimator with an $\ell_1$-loss function. The intuition behind the advantage of the LASSO-type estimator is that the estimator is robust to "sparse" attacks. In our work, the sparsity of attacks is defined through probability $p$ (Definition 1). Suppose that the attack vectors $\bar{d}_t$ are nonzero and large for a small number of time indices $t$. The least-squares estimator (2) will generally not be able to *exactly* recover the underlying parameter $A$ since the loss function is smooth with respect to the error term $x_{t+1} - Af(x_t)$. In contrast, we prove that the LASSO-type estimator (3) is able to *exactly* recover the parameter $A$ given a sufficiently large number of time steps. The exact recovery is guaranteed through the nonsmooth loss function, which prefers solutions that result in a small number of nonzero error terms $x_{t+1} - Af(x_t)$. This property of the LASSO-type estimator (3) fits better to our setting than the least-squares estimator (2).

In this work, the goal is to prove the efficacy of the above estimator by obtaining mild conditions under which the ground truth $\bar{A}$ can be *exactly recovered* by the estimator (3). More specifically, we seek to address the following key questions:

i) What are the *necessary and sufficient* conditions under which $\bar{A}$ is an optimal solution to the optimization problem (3) or the unique optimal solution?

ii) What is the minimum number of samples required to ensure that the necessary and sufficient conditions for exact recovery hold with high probability under certain assumptions?

In this work, we provide answers to the above questions. Section 2 presents a comprehensive literature review on system identification and robust estimation problems. In Section 3, we analyze the necessary and sufficient conditions for the global optimality of $\bar{A}$ for the problem (3). Furthermore, we establish conditions under which $\bar{A}$ is the unique solution, thereby addressing Question (i). Next, in Sections 5 and 6, we derive lower bounds on the number of samples $T$ required to guarantee that $\bar{A}$ is the unique solution with high probability. We consider two distinct cases: when the basis function $f$ is bounded when it is Lipschitz continuous. These results provide an answer to Question (ii). In Section 7, we introduce a faster learning strategy when nonzero disturbances fail to sufficiently explore the system state space. Finally, in Section 8, we present numerical experiments that support our theoretical findings. This work constitutes the first non-asymptotic sample complexity analysis for exact recovery in the nonlinear system identification problem.

**Notation.** For a positive integer $n$, we use $\mathbf{0}_n$ and $\boldsymbol{I}_n$ to denote the $n$-dimensional vector with all entries being 0 and the $n$-by-$n$ identity matrix. For a matrix $Z$, $\|Z\|_F$ denotes its Frobenius norm and $\mathbb{S}_F$ is the unit sphere of matrices with Frobenius norm $\|Z\|_F = 1$. For two matrices $Z_1$ and $Z_2$, we use $\langle Z_1, Z_2 \rangle = \text{Tr}(Z_1^\top Z_2)$ to denote the inner-product. For a vector $z$, $\|z\|_2$ and $\|z\|_\infty$ denote its $\ell_2$- and $\ell_\infty$-norms, respectively. Moreover, $\mathbb{S}^{n-1}$ is the unit ball $\{z \in \mathbb{R}^n | \|z\|_2 = 1\}$. Given two functions $f$ and $g$, the notation $f(x) = \Theta[g(x)]$ and $f(x) = \Omega[g(x)]$ means that there exist universal positive constants $c_1$ and $c_2$ such that $c_1 g(x) \leq f(x) \leq c_2 g(x)$ and $f(x) \geq c_1 g(x)$, respectively. The relation $f(x) \lesssim g(x)$ holds if there exists a universal positive constant $c_3$ such that $f(x) \leq c_3 g(x)$ holds with high probability when $T$ is large. The relation $f(x) \gtrsim g(x)$ holds if $g(x) \lesssim f(x)$. $|S|$ shows the cardinality of a given set $S$. $\mathbb{P}(\cdot)$ and $\mathbb{E}(\cdot)$ denote the probability of an event and the expectation of a random variable. A Gaussian random vector $X$ with mean $\mu$ and covariance matrix $\Sigma$ is written as $X \sim \mathcal{N}(\mu, \Sigma)$.

## 2 Literature Overview

Until recently, research on the system identification problem primarily focused on the asymptotic properties of the least squares estimator (LSE) (Chen & Guo, 2012; Ljung et al., 1999; Ljung & Wahlberg, 1992;

Bauer et al., 1999). However, with the increasing prominence of statistical learning theory (Vershynin, 2018; Wainwright, 2019), understanding the number of samples required to achieve a given error threshold in system identification has become a topic of significant interest. For a comprehensive overview of existing results and proof techniques, we refer the reader to the survey by Tsiamis et al. (2023). The non-asymptotic analysis of system identification has primarily focused on linear time-invariant (LTI) systems under the assumption of i.i.d. noise. Early research in this area relied on mixing arguments, which heavily depended on system stability (Kuznetsov & Mohri, 2017; Rostamizadeh & Mohri, 2007). More recent studies have employed martingale and small-ball techniques to establish sample complexity guarantees for least squares estimators applied to LTI systems (Simchowitz et al., 2018; Faradonbeh et al., 2018; Tsiamis & Pappas, 2019). These works demonstrate that the LSE converges to the true system parameters at a rate of $T^{-1/2}$, where $T$ denotes the number of samples. Furthermore, these results have been applied to the linear-quadratic regulator problem, where adaptive control techniques leverage system identification results to achieve optimal regret bounds (Dean et al., 2020; Abbasi-Yadkori & Szepesvári, 2011; Dean et al., 2019).

The nonlinear system identification problem has been extensively studied (Noël & Kerschen, 2017; Nowak, 2002). However, research on the non-asymptotic analysis of nonlinear system identification remains in its early stages and has primarily focused on parameterized nonlinear systems. Recursive and gradient-based algorithms designed for the least squares loss function have been shown to asymptotically converge to the true system parameters at a rate of $T^{-1/2}$ for nonlinear systems with a known link function $\phi$ of the form $\phi(\bar{A}x_t)$, using martingale techniques (Foster et al., 2020) and mixing time arguments (Sattar & Oymak, 2022). More recently, Ziemann et al. (2022) established sample complexity guarantees for nonparametric learning of nonlinear system dynamics, demonstrating a convergence rate that scales as $T^{-1/(2+q)}$, where $q$ depends on the complexity of the function class in which the true dynamics reside. Notably, existing studies on both linear and nonlinear system identification have largely assumed i.i.d. (sub-)Gaussian noise structures, limiting their applicability to more general disturbance models.

Despite the growing interest in non-asymptotic system identification, research on system identification using non-smooth estimators capable of handling dependent and adversarial noise vectors remains limited to linear systems. In addition, the main focus of aforementioned works is on the non-asymptotic approximate recovery of the underlying parameter $A$. Under safety-critical constraints, it is ideal to achieve the non-asymptotic exact recovery of the parameter $A$, possibly with additional assumptions on the adversarial noise. There exist early attempts on the exact recovery guarantees. Feng & Lavaei (2021) and Feng et al. (2023) investigated a non-smooth convex estimator in the form of the least absolute deviation estimator, analyzing the conditions required for exact recovery of system dynamics using Karush-Kuhn-Tucker (KKT) conditions and the Null Space Property from the LASSO literature. Subsequently, Yalcin et al. (2023) demonstrated that exact recovery of system parameters is achievable with high probability, even when more than half of the data is corrupted, opening new avenues for adversarially robust system identification. Compared to Yalcin et al. (2023), the presence of nonlinear basis functions in our setting makes it impossible to analyze the optimization problem by explicitly expressing $x_t$; see the proof of Theorem 2 in Yalcin et al. (2023). Note that when the system is in the form of $x_{t+1} = Ax_t$, then $x_t$ can be written directly as $A^t x_0$, and we only need to analyze the eigenvalues of $A$. For a nonlinear system in the form of $x_{t+1} = f(x_t)$, writing $x_t$ in terms of $x_0$ needs the composition of $t$ functions, and this cannot be done analytically. Unlike linear systems, there is no direct counterpart to eigenvalue analysis for nonlinear systems. This fundamental challenge is widely acknowledged in nonlinear systems literature within control theory, and as a result, many results known for linear systems do not extend to the nonlinear setting. Consequently, our proof for the bounded case is novel and fundamentally different from the approach in Yalcin et al. (2023). Finally, by leveraging the generalized Farkas' lemma, we establish necessary and sufficient conditions in Section 3 that are both novel and stronger than the sufficient conditions provided in Yalcin et al. (2023).

On the other hand, robust regression techniques have been developed by incorporating regularizers into the objective function (Xu et al., 2009; Bertsimas & Copenhaver, 2018; Huang et al., 2016). Additionally, the robust estimation literature has introduced several non-smooth estimators, including M-estimators, least absolute deviation estimators, convex estimators, least median squares, and least trimmed squares (Seber & Lee, 2012). The convex estimator in (3) was proposed in Bako & Ohlsson (2016); Bako (2017) in the context of robust regression, demonstrating that exact recovery is achievable given an infinite number of samples.

However, these studies lack a non-asymptotic analysis of sample complexity. Furthermore, their analytical techniques are not directly applicable to dynamical systems due to the presence of autocorrelation among samples.

Recent works (Wu et al., 2022; Kumar et al., 2022) have focused on reinforcement learning (RL), where the primary objective is to maximize a reward function. In contrast, system identification aims to recover the underlying system dynamics, and in many applications, a naturally defined reward function may not exist. Moreover, both of these RL studies assume that perturbations are bounded, a restrictive assumption that may not hold in real-world scenarios. More importantly, attempting to control a system without first learning its dynamics (e.g., using model-free RL techniques) poses significant risks. During exploration, an inadequate policy could shift the system state beyond safe operational limits, potentially leading to instability; see the survey by Moerland et al. (2023). For safety-critical systems, it is typically necessary to first learn the system dynamics before applying a control strategy, whether it be a classical optimal control method or an RL-based approach. Our work focuses on learning the system model in the presence of adversaries on its dynamics. Existing RL approaches, including those in Wu et al. (2022); Kumar et al. (2022), address a fundamentally different problem. Additionally, while the field of robust model-based RL is well-developed, our setting involves unknown system dynamics, necessitating the use of model-free RL techniques.

## 3 Global Optimality of Ground Truth and Uniqueness of Global Solutions

In this section, we derive conditions under which the ground truth $\bar{A}$ is a global minimizer to the optimization problem (3). Given the system dynamics, this optimization problem can be reformulated as

$$\min_{A \in \mathbb{R}^{n \times m}} \sum_{t=0}^{T-1} \|(\bar{A} - A)f(x_t) + \bar{d}_t\|_2, \tag{4}$$

where $x_0, \ldots, x_T$ are generated according to the unknown system under disturbances. To facilitate the analysis, we define the set of time instances where disturbances are nonzero as $\mathcal{K} := \{t \mid \bar{d}_t \neq 0\}$ and introduce the normalized disturbances as

$$\hat{d}_t := \bar{d}_t / \|\bar{d}_t\|_2, \quad \forall t \in \mathcal{K}.$$

The following theorem provides a necessary and sufficient condition for $\bar{A}$ to be a global minimizer of the problem in (4).

**Theorem 1** (Necessary and sufficient condition for optimality). *The ground truth matrix $\bar{A}$ is a global solution to problem (4) if and only if*

$$\sum_{t \in \mathcal{K}} \hat{d}_t^\top Z f(x_t) \leq \sum_{t \in \mathcal{K}^c} \|Z f(x_t)\|_2, \quad \forall Z \in \mathbb{R}^{n \times m}, \tag{5}$$

*where $\mathcal{K}^c := \{0, \ldots, T-1\} \backslash \mathcal{K}$ denotes the set of time indices where disturbances are zero.*

Theorem 1 provides a necessary and sufficient condition for the well-specifiedness of the optimization problem in (4). Intuitively, the left-hand side of (5) represents the impact of nonzero disturbances, while the right-hand side corresponds to the standard system dynamics. If the disturbances do not dominate the correct system dynamics, then the estimator can successfully recover the ground truth dynamics. The condition in (5) is derived using the generalized Farkas' lemma, which eliminates the need for inner approximation of the $\ell_2$-ball by an $\ell_\infty$-ball, as in Yalcin et al. (2023). Consequently, the sample complexity bounds obtained in this work are stronger than those in Yalcin et al. (2023) when applied to the special case of linear systems. Further details on these bounds are provided in Sections 5 and 6.

Using the condition established in Theorem 1, we derive both sufficient and necessary conditions for the optimality of $\bar{A}$ in Corollaries 1 and 2.

**Corollary 1** (Sufficient condition for optimality). *If it holds that*

$$\sum_{t \in \mathcal{K}} \|Z f(x_t)\|_2 \leq \sum_{t \in \mathcal{K}^c} \|Z f(x_t)\|_2, \quad \forall Z \in \mathbb{R}^{n \times m}, \tag{6}$$

*then the ground truth matrix $\bar{A}$ is a global solution to problem (4).*

**Corollary 2** (Necessary condition for optimality)**.** *If the ground truth matrix $\bar{A}$ is a global solution to problem (4), then it holds that*

$$\left\|\sum_{t\in\mathcal{K}} f(x_t)\hat{d}_t^\top\right\|_F \le \sum_{t\in\mathcal{K}^c} \|f(x_t)\|_2. \tag{7}$$

*In the case when $m = 1$, condition (7) is necessary and sufficient.*

The proof of Corollary 1 is a direct application of the Cauchy-Schwartz inequality and we omit the proof. The proof of Corollary 2 is provided in Appendix B. The conditions established above are more general than many existing results in the literature; see Examples 1 and 2 in Appendix A for further discussion.

Next, we derive conditions under which the ground truth matrix $\bar{A}$ is the unique solution to the optimization problem in (4). We present the following necessary and sufficient conditions for the uniqueness of global solutions, which extends the result of Theorem 1.

**Theorem 2** (Necessary and sufficient condition for uniqueness)**.** *Suppose that condition (5) holds. The ground truth $\bar{A}$ is the unique global solution to problem (4) if and only if the following logical condition holds for every nonzero $Z \in \mathbb{R}^{n\times m}$:*

$$\sum_{t\in\mathcal{K}} \hat{d}_t^\top Z f(x_t) = \sum_{t\in\mathcal{K}^c} \|Zf(x_t)\|_2 \implies \sum_{t\in\mathcal{K}} \left|\hat{d}_t^\top Z f(x_t)\right| < \sum_{t\in\mathcal{K}} \|Zf(x_t)\|_2, \tag{8}$$

*which means that whenever the left-hand side equality is satisfied for some nonzero $Z$, the right-hand side inequality must also hold.*

Based on Theorem 2, the following corollary provides a sufficient condition for the uniqueness of $\bar{A}$, which is more practical to verify compared to condition (8). Notably, this corollary also generalizes the sufficiency part of Corollary 2 to the multi-dimensional setting.

**Corollary 3** (Sufficient condition for uniqueness)**.** *If it holds that*

$$\sum_{t\in\mathcal{K}} \hat{d}_t^\top Z f(x_t) < \sum_{t\in\mathcal{K}^c} \|Zf(x_t)\|_2, \quad \forall Z \in \mathbb{R}^{n\times m} \quad \text{s.t. } Z \ne 0, \tag{9}$$

*then the ground truth matrix $\bar{A}$ is the unique global solution to problem (4).*

*Proof.* The logical condition in (8) states that whenever the left-hand side equality is satisfied for some nonzero $Z$, the right-hand side inequality must also hold. Under the assumption in (9), there is no nonzero $Z$ satisfying the left-hand-side equality. This implies that the logical condition in (8) automatically holds and, thus, Theorem 2 implies the uniqueness of $\bar{A}$. $\square$

Theorem 2 strengthens and generalizes existing results for first-order systems, specifically Theorem 1 in Feng & Lavaei (2021). For further discussion, see Example 3 in Appendix A.

## 4   Disturbance Model and Semi-Oblivious Attacks

To model sparse disturbances, we consider the frequency at which the system experiences a nonzero disturbance. This is captured by the sparsity probability $p$, defined as follows:

**Definition 1** (Probabilistic sparsity model)**.** *For each time instance $t$, the disturbance vector $\bar{d}_t$ is nonzero with probability $p \in (0, 1)$. Furthermore, the occurrences of disturbances are independent across time instances.*

Existing results have predominantly focused on the case where $p = 1$, under which the system dynamics cannot be learned exactly in finite time. To illustrate the impact of sparsity, consider the scenario where $\bar{d}_t$

follows a Gaussian distribution for all $t \in \mathcal{K}$ and independent of previous disturbances $\bar{d}_1, ..., \bar{d}_{t-1}$. When $p = 1$, it follows that there exists no finite time $T$ for which the correct dynamics matrix $\bar{A}$ is a solution to the least-square estimator (2) almost surely. As an alternative to the exact recovery, existing works have focused on the finite-time approximate recovery of parameter $A$ and established the convergence of the estimation error (Noël & Kerschen, 2017; Nowak, 2002; Foster et al., 2020; Sattar & Oymak, 2022; Ziemann et al., 2022). See our discussion on the approximate recovery results in Section 2. However, the exactly recovery of the underlying system parameters is possible under suitable assumptions (e.g., Definition 1) and is preferred since it provides a stronger recovery guarantee against adversarial attacks.

Moreover, when $p < 1$, the LSE also fails to achieve exact recovery even under an i.i.d zero-mean Gaussian disturbance structure. To illustrate the failure of LSE, we define the following matrices:

$$F_T = \begin{bmatrix} f(x_0) & f(x_1) & \cdots & f(x_{T-1}) \end{bmatrix} \text{ and } X_T = \begin{bmatrix} x_1 & x_2 & \cdots & x_T \end{bmatrix},$$

where $F_T, X_T \in \mathbb{R}^{n \times T}$. In addition, we define the disturbance matrix $\bar{D}_T = \begin{bmatrix} \bar{d}_0 & \bar{d}_1 & \cdots & \bar{d}_{T-1} \end{bmatrix}$, where $\bar{D}_T \in \mathbb{R}^{n \times T}$. Under these definitions, the system updates can be expressed as $X_T = \bar{A}F_T + \bar{D}_T$. Thus, the closed-form solution of the LSE is given by $\hat{A} = (F_T F_T^\top)^{-1} F_T X_T^\top$, and the estimation error can be written as $\hat{A} - \bar{A} = (F_T F_T^\top)^{-1} F_T \bar{D}_T^\top$. When $T$ is sufficiently large, the matrices $F_T$ and $\bar{D}_T$ become full-rank; consequently, the LSE estimation error, $\hat{A} - \bar{A}$, never attains zero error. In contrast, the results in Sections 5 and 6 conclude that for every $p \in (0, 1)$, the ground truth matrix $\bar{A}$ becomes the unique solution of the non-smooth estimator (3), provided that $T$ exceeds a certain threshold. This result represents a highly specialized case of the broader findings presented in this paper and highlights the advantage of incorporating zero disturbances. Specifically, having some instances of zero disturbances enables a transition from asymptotic learning to finite-time learning, demonstrating the efficacy of the proposed approach.

It is important to note that in this work, the disturbance vectors $\bar{d}_t$'s are allowed to be correlated over time. Definition 1 is only about the times at which disturbances are nonzero. Additionally, we do not assume that the sparsity probability $p$ or the model generating the disturbances is known. Recalling the definition $\mathcal{K} := \{t \mid \bar{d}_t \neq 0\}$, we observe that with probability at least $1 - \exp[-\Theta(pT)]$, the cardinality of $\mathcal{K}$ satisfies $|\mathcal{K}| = \Theta(pT)$. When $p$ is close to 0, the system is rarely affected by disturbances. The absence of disturbances, however, may lessen the rate of exact recovery due to insufficient exploration of the system space. A potential approach to mitigate this issue is outlined in Section 7. Nevertheless, this paper primarily focuses on the case where $p$ is close to 1, meaning that the system experiences disturbances at nearly all time steps.

## 4.1 Non-degenerate Condition

In the following theorem, we demonstrate that in the absence of assumptions on the disturbance vectors, no estimator can reliably learn the system dynamics.

**Theorem 3.** *Consider the linear system $x_{t+1} = \bar{A}x_t + \bar{d}_t, x_0 = \mathbf{0}_n$, together with a linear subspace $D \subset \mathbb{R}^n$ whose dimension is less than rank of $\bar{A}$. Assume that the disturbance $\bar{d}_t$ is always chosen from this not-full-dimensional subspace. Then, there does not exist any estimator of the form*

$$\hat{A}_T \in \arg\min_{A \in \mathbb{R}^{n \times n}} g(A; \{x_t\}_{t=0}^T)$$

*that uniquely recovers the matrix $\bar{A}$ all the time from the states $x_0, ..., x_T$ no matter how large $T$ is. Here, $g(A; \{x_t\}_{t=0}^T)$ is the function of $A$ that depends on the system states over time.*

To avoid the degenerate case, we assume that the norm of basis function is lower-bounded under conditional expectation after a nonzero disturbance.

**Assumption 1** (Non-degenerate condition). *Conditional on the past information $\mathcal{F}_t$ and the event that $\bar{d}_t \neq \mathbf{0}_n$, the disturbance vector and the basis function satisfy*

$$\lambda_{min} \left[ \mathbb{E} \left[ f(x + \bar{d}_t) f(x + \bar{d}_t)^\top \mid \mathcal{F}_t, \bar{d}_t \neq \mathbf{0}_n \right] \right] \geq \lambda^2, \quad \forall x \in \mathbb{R}^n,$$

*where $\lambda_{min}(F)$ is the minimal eigenvalue of matrix $F$ and $\lambda > 0$ is a constant.*

Intuitively, the non-degenerate assumption ensures sufficient exploration of the trajectory in the state space. More specifically, for the condition in (9) to hold, it is necessary that the matrix

$$[f(x_t), \ t \in \mathcal{K}^c] \in \mathbb{R}^{m \times (T - |\mathcal{K}|)} \tag{10}$$

has rank-$m$; see the proof of Theorem 5 for further details. The non-degenerate assumption guarantees that the basis function evaluations $f(x + \bar{d}_t)$ spans the entire state space in expectation. As a result, the matrix in (10) is full-rank with high probability when the number of samples $T$ is sufficiently large.

### 4.2 Semi-oblivious Condition

Consider the scenario where the disturbances affecting the system are adversarially designed. In this case, we refer to these disturbances as attacks, with $p$ representing the frequency at which the system is under attack. Two key problems arise in the context of attack analysis: *(1) attack problem* where the adversary aims to design the attack vectors $\bar{d}_t$ to maximize the disruption to the system, and *(2) defense problem* where the system operator seeks to detect any suspicious attack and nullify it. Two common strategies often used in combination to design an effective defense mechanism are (i) inspecting the system inputs to detect anomalies indicative of attacks and (ii) analyzing collected state values to infer whether an attack has occurred. Based on Theorem 3, if the attacker has complete freedom in selecting attack values, the most effective strategy is to constrain them to a specific subspace where no estimator can function reliably. However, such biased attack strategies are easier to detect using defense strategy (i), as the operator can analyze the statistical behavior of the inputs and identify deviations from natural disturbances, flagging them as attacks. Thus, the risk for an attacker employing extreme attacks is that they increase the likelihood of detection and nullification by a well-designed defense mechanism. Consequently, a strategic adversary should avoid highly conspicuous attack patterns and instead focus on attacks that have a lower probability of detection.

As an example, consider the first-order system $x_{t+1} = x_t + \bar{d}_t$ where $\bar{d}_t \in \mathbb{R}$. Suppose this represents a physical system that cannot accept inputs exceeding a given magnitude limit $\gamma$. The most severe attack in this case would be to set $\bar{d}_t$ to be equal to $\gamma$ at all times, driving the state to grow as rapidly as possible. However, a well-designed defense mechanism could quickly detect this attack by recognizing that the injected input is not a natural disturbance but rather a deliberately crafted adversarial input. To evade detection, the attacker must adopt a more subtle approach. Instead of consistently applying the maximum allowable input, the attacker may alternate between positive and negative disturbances, choosing $\bar{d}_t$ to be $+\bar{\gamma}$ and $-\bar{\gamma}$ for some $\bar{\gamma} < \gamma$. This strategy achieves two key objectives: (i) avoiding hitting the maximum limit and (ii) making the attack look like a disturbance by having a zero mean. Although this alternating attack pattern is less detectable, it still has a significant impact on the system. Specifically, it increases the variance of $x_t$, causing oscillatory behavior that can degrade system performance. This example illustrates a broader principle: an effective attack strategy should involve alternating positive and negative perturbations rather than consistently applying unidirectional inputs. To formally capture this concept, we introduce the notion of attack/disturbance direction and define a semi-oblivious condition in the attack/disturbance process. This condition aligns with existing robustness criteria in the literature (Candès et al., 2011; Chen et al., 2021). To do so, we define the filtration $\mathcal{F}_t := \sigma\{x_0, x_1, \ldots, x_t\}$.

**Assumption 2** (Semi-oblivious condition). *Conditional on the past information $\mathcal{F}_t$ and the event that $\bar{d}_t \neq \mathbf{0}_n$, the disturbance direction $\hat{d}_t = \bar{d}_t / \|\bar{d}_t\|_2$ is zero-mean.*

The semi-oblivious condition has been widely studied in real-world systems, particularly in the context of stealthy attacks. To illustrate this concept, consider another example from an energy system comprising two nodes: Node 1 represents a neighborhood of homes, and Node 2 is a power supplier owned by a utility company. Every five minutes, Node 1 reports to Node 2 the amount of electrical power required for the next five-minute period. Suppose the neighborhood requires a constant power demand of 1 unit for the next five hours. However, an attacker has compromised the communication channel between Node 1 and Node 2, altering the reported demand from 1 unit to either $1 - e$ or $1 + e$ every half hour, where $e$ is a large perturbation relative to 1 unit. Since the average of $1 - e$ and $1 + e$ is still 1, this could serve as a semi-oblivious attack. However, this manipulation disrupts the power balance: when Node 1 actually requires 1 unit of power, but Node 2 instead generates either $1 - e$ or $1 + e$, the resulting mismatch violates

the fundamental physical constraints of the grid. Such discrepancies can trigger grid instability, potentially leading to large-scale blackouts. Semi-oblivious attacks of this kind have been observed in various parts of the world, resulting in severe power outages. The cyberattack problem in power systems aligns closely with our mathematical models. Notably, power system operators employ hypothesis testing techniques to detect anomalies in reported data. If the injected disturbances exhibit a nonzero mean, they would be flagged as suspicious. However, by maintaining a zero mean, semi-oblivious evade detection while still destabilizing the system.

Due to the poor performance of LSE under the disturbance structure characterized by the semi-oblivious condition and the probabilistic sparsity model, our work focuses on analyzing the LASSO-type estimator and understanding the conditions under which exact recovery is attainable in such settings. We emphasize that this disturbance structure encompasses not only some sparse disturbances but also certain types of attacks. For instance, the disturbance structure may consist of a combination of sparse, zero-mean Gaussian input vectors, denoted as $\bar{e}_t$, and semi-oblivious attack vectors, $\bar{f}_t$, both of which satisfy the required assumptions. Suppose the $\bar{e}_t$ and $\bar{f}_t$ follow the probabilistic sparsity model with probabilities $p'$ and $p$, respectively. In that case, the overall disturbance, given by $\bar{d}_t = \bar{e}_t + \bar{f}_t$, is also semi-oblivious and follows the probabilistic sparsity model with probability at most $p' + p < 1$.

In the following two sections, we provide lower bounds on the sample complexity $T$ such that the ground truth $\bar{A}$ is the unique solution to problem (4). The following section derives results under Assumption 2. Following that, we extend our analysis to unbounded basis functions, which require Assumption 6. While Assumption 6 is somewhat stronger than Assumption 2, it remains a practical assumption. Note that the former assumption requires the disturbance directions to have a zero meanwhile the latter assumption requires the disturbance directions to follow a uniform distribution. In both cases, the magnitudes of the nonzero disturbances/attacks remain unrestricted, meaning that adversaries can inject disturbances of arbitrary values to maximize their impact on the system.

## 5 Bounded Basis Function

In this section, we analyze the case where the basis function $f$ is bounded.

**Assumption 3** (Bounded basis function). *The basis function $f : \mathbb{R}^n \mapsto \mathbb{R}^m$ satisfies*

$$\|f(x)\|_\infty \leq B, \quad \forall x \in \mathbb{R}^n,$$

*where $B > 0$ is a constant.*

The following theorem establishes that when the sample complexity is sufficiently high, the estimator in (3) achieves exact recovery of the ground truth matrix $\bar{A}$ with high probability.

**Theorem 4** (Exact recovery for bounded basis function). *Suppose that Assumptions 2-1 hold and the disturbance vector $\bar{d}_t$ is nonzero with probability $p \in (0,1)$. Define $\kappa := B/\lambda \geq 1$. For all $\delta \in (0,1]$, if the sample complexity $T$ satisfies*

$$T = \Omega\left[\frac{m^2\kappa^4}{p(1-p)^2}\left[mn\log\left(\frac{m\kappa}{p(1-p)}\right) + \log\left(\frac{1}{\delta}\right)\right]\right], \tag{11}$$

*then $\bar{A}$ is the unique global solution to problem (4) with probability at least $1 - \delta$.*

The above theorem provides a non-asymptotic bound on the sample complexity required for exact recovery with a specified probability of at least $1 - \delta$. The lower bound scales as $m^3n$, indicating that the required number of samples increases as the number of states $n$ and the number of basis functions $m$ grow. In addition, the sample complexity increases when $B$ is larger or $\lambda$ is smaller. This aligns with the intuition that $B$ reflects the size of the space spanned by the basis function, and $\lambda$ measures the *speed* of exploring the spanned space.

Regarding the dependence on the sparsity probability $p$, the following theorem demonstrates that the scaling factor $1/[p(1-p)]$ is unavoidable under the probabilistic sparsity model. Furthermore, the theorem also establishes a lower bound on the sample complexity that depends on $m$ and $\log(1/\delta)$.

**Theorem 5.** *Suppose that the disturbance vector $\bar{d}_t$ is nonzero with probability $p \in (0, 1)$ and the sample complexity satisfies*

$$T < \frac{m}{2p(1-p)}.$$

*Then, there exists a basis function $f : \mathbb{R}^n \mapsto \mathbb{R}^m$ and a disturbance model such that Assumptions 2-1 hold and the global solutions to problem (4) are not unique with probability at least $\max\left\{1 - 2\exp\left(-m/3\right), 2[p(1-p)]^{T/2}\right\}$. Furthermore, given a constant $\delta \in (0, 1]$, if*

$$T < \max\left\{\frac{m}{2p(1-p)}, \frac{2}{-\log[p(1-p)]}\log\left(\frac{2}{\delta}\right)\right\},$$

*then the global solutions to problem (4) are not unique with probability at least $\max\left\{1 - 2\exp\left(-m/3\right), \delta\right\}$.*

We note that the result of Theorem 5 hold for both bounded and Lipschitz base functions, which are studied in Section 6.

**Remark 1.** *A key objective of this paper is to show that the exact recovery in finite time is achievable when $p$ is close to 1, indicating that the system operates under semi-oblivious attacks most of the time. In Theorem 4, we establish an upper bound on the required time horizon for exact recovery, a result with significant implications for real-world systems. Conversely, the lower bound derived in Theorem 5 serves primarily as a theoretical result. Unlike large-scale machine learning problems, where the problem size can reach tens of millions, real-world dynamical systems typically have far fewer states, often fewer than several thousand. As a result, our upper bound already provides a practical estimate of the required sample complexity. While further tightening the lower bound remains a relevant and theoretically interesting problem, its practical impact is likely to be marginal, given that the current upper bound is already within a feasible range for real-world applications.*

## 6 Lipschitz Basis Function

In this section, we consider the case when the basis function $f(x)$ is Lipschitz continuous in $x$. Specifically, we make the following assumption.

**Assumption 4** (Lipschitz basis function)**.** *The basis function $f : \mathbb{R}^n \mapsto \mathbb{R}^m$ satisfies*

$$f(\mathbf{0}_n) = \mathbf{0}_m \quad and \quad \|f(x) - f(y)\|_2 \le L\|x - y\|_2, \quad \forall x, y \in \mathbb{R}^n,$$

*where $L > 0$ is the Lipschitz constant.*

As a special case of Assumption 4, the basis function of a linear system is $f(x) = x$, which satisfies the Lipschitz condition with $L = 1$.

**Remark 2.** *It is important to note that the assumptions of boundedness or Lipschitz continuity are always satisfied in dynamical systems, as the user has the flexibility to select appropriate basis functions to ensure these properties. Specifically, the user can choose an arbitrary set of basis functions to approximate the unknown function as a linear combination of these bases. This setting differs from classical machine learning problems, where the model is trained to learn an unknown function, and there is no direct control over its Lipschitz continuity. Conversely, in dynamical systems, the user can impose constraints on the basis functions to ensure desirable mathematical properties. However, if unbounded basis functions or functions with high Lipschitz constants are restricted, a larger number of basis functions may be required to achieve an accurate approximation of the unknown function. Moreover, many real-world dynamical systems, ranging from robotics to energy systems, inherently exhibit smooth and well-behaved dynamics due to their foundation in physical laws, such as Newtonian mechanics and Kirchhoff's circuit laws. This stands in contrast to various machine learning problems, where the optimal policy may be inherently non-smooth and highly complex, making function approximation significantly more challenging.*

Additionally, we assume that the spectral norm of $\bar{A}$ is bounded to ensure system stability.

**Assumption 5** (System stability)**.** *The ground truth $\bar{A}$ satisfies*

$$\rho := \left\| \bar{A} \right\|_2 < \frac{1}{L}.$$

Assumption 5 is related to the asymptotic stability of the dynamical system and is sufficient to prevent the finite-time divergence of the system trajectories. In Theorem 7, we demonstrate that this stability condition may also be necessary for exact recovery. Finally, we make the additional assumption that each disturbance follows a sub-Gaussian distribution.

**Assumption 6** (Sub-Gaussian disturbances)**.** *Conditional on the filtration $\mathcal{F}_t$ and the event that $\bar{d}_t \neq \mathbf{0}_n$, the disturbance vector $\bar{d}_t$ is expressed as the product $\ell_t \hat{d}_t$, where*

1. *$\hat{d}_t \in \mathbb{R}^n$ and $\ell_t \in \mathbb{R}$ are independent conditional on $\mathcal{F}_t$ and $\bar{d}_t \neq \mathbf{0}_n$;*

2. *$\hat{d}_t$ is a zero-mean unit vector, namely, $\mathbb{E}(\hat{d}_t \mid \mathcal{F}_t, \bar{d}_t \neq \mathbf{0}_n) = \mathbf{0}_n$ and $\|\hat{d}_t\|_2 = 1$;*

3. *$\ell_t$ is zero-mean and sub-Gaussian with parameter $\sigma$.*

As a special case, the sub-Gaussian assumption is guaranteed to hold if there is an upper bound on the magnitude of each disturbance. To be more specific, suppose that $\|\bar{d}_t\|_2 \leq B$ for all $t$. In this case, we have $\ell_t \leq B$ for all $t$ and thus, the magnitude $\ell_t$ is sub-Gaussian with parameter $\sigma = B$. The bounded disturbance case is common in practical applications since real-world systems do not accept inputs that are arbitrarily large. For example, physical devices have a clear limitation on the input size, and the disturbances/attacks cannot exceed that limit. In Assumption 6, the components $\hat{d}_t$ and $\ell_t$ respectively represent the direction and intensity (such as magnitude) of each attack/disturbance. While the intensity parameters $\ell_t$'s could be correlated over time, both $\hat{d}_t$ and $\ell_t$ are assumed to be zero-mean, aligning with the semi-oblivious condition stated before.

Under these assumptions, we can establish that high-probability exact recovery is achievable, provided that the sample size $T$ is sufficiently large.

**Theorem 6** (Exact recovery for Lipschitz basis function)**.** *Suppose that Assumptions 1-6 hold and the disturbance vector $\bar{d}_t$ is nonzero with probability $p \in (0, 1)$. Define $\kappa := \sigma L / \lambda \geq 1$. If the sample complexity $T$ satisfies*

$$T = \Omega \left[ \max \left\{ \frac{\kappa^{10}}{(1 - \rho L)^3 (1 - p)^2}, \frac{\kappa^4}{p(1 - p)} \right\} \times \left[ mn \log \left( \frac{1}{(1 - \rho L)\kappa p(1 - p)} \right) + \log \left( \frac{1}{\delta} \right) \right] \right], \qquad (12)$$

*then $\bar{A}$ is the unique global solution to problem (4) with probability at least $1 - \delta$.*

Theorem 6 provides a non-asymptotic sample complexity bound for the case where the basis function is Lipschitz continuous. As a special case, when the basis function is linear, i.e., $f(x) = x$, and the attack/disturbance vector $\bar{d}_t$ follows the Gaussian distribution $\mathcal{N}(\mathbf{0}_n, \sigma^2 \mathbf{I}_n)$ conditional on $\mathcal{F}_t$, we have $\kappa = 1$. Compared to Theorem 4, the dependence on the nonzero disturbance probability $p$ is improved from $1/[p(1 - p)^2]$ to $1/[p(1 - p)]$, a consequence of the stability condition (Assumption 5). In addition, the dependence on the dimension $m$ is improved from $m^3$ to $m$. Intuitively, the improvement is achieved by improving the upper bound on the norm $\|f(x_t)\|_2$. In the bounded basis function case, the norm is bounded by $\sqrt{m}B$, while in the Lipschitz basis function case, the norm is bounded by $\sigma L$ with high probability, which is independent of the dimension $m$. Finally, the sample complexity bound grows with the parameter $\kappa = \sigma L / \lambda$ and the gap $1 - \rho L$, which is also consistent with the intuition.

Conversely, we can construct counterexamples demonstrating that when the stability condition (Assumption 5) is violated, exact recovery fails with probability at least $p$.

**Theorem 7** (Failure of exact recovery for unstable systems)**.** *There exists a system such that Assumptions 1, 4 and 6 are satisfied, but for all $T \geq 1$, the ground truth $\bar{A}$ is not a global solution to problem (4) with probability at least $p[1 - (1 - p)^{T-1}]$.*

# 7 Absence of Semi-Oblivious Attacks

In Sections 5 and 6, we derived the required number of samples for exact recovery in the presence of sparse and semi-oblivious disturbances using bounded and Lipschitz basis functions. For bounded basis functions, the sample complexity scales as $p^{-1}(1-p)^{-2}$ in terms of the nonzero disturbance probability, up to a logarithm factor, as stated in Theorem 4. Similarly, Theorem 6 established that, for Lipschitz basis functions, the sample complexity scales as $\max\{(1-p)^{-2}, p^{-1}(1-p)^{-1}\}$ with respect to the nonzero disturbance probability. Although the theoretical results in these sections are quite promising whenever nonzero disturbance probability $p > 0.5$, it may seem counterintuitive that the sample complexity increases as $p$ decreases and approaches zero, given that a smaller $p$ implies fewer nonzero disturbances and attack injections into the system. This phenomenon can be explained through the concept of exploration. When $p$ is small, the lack of disturbances reduces the rate at which the system state space is explored, thereby increasing the number of samples required for exact recovery. Numerical experiments further support this theoretical observation, the results of which are presented in Appendix C.

It is well-known that random excitation signals into a dynamical system can accelerate convergence in system identification problems. To leverage the sample complexity bounds derived in earlier sections and enhance the learning rate, we define the disturbance vector $\bar{d}_t$ as a combination of the random input vectors injected by the controller, $\bar{e}_t$, and the semi-oblivious attacks designed by an adversarial agent, $\bar{f}_t$, such that $\bar{d}_t = \bar{e}_t + \bar{f}_t$. In the absence of random excitation input injections from the controller, we have $\bar{d}_t = \bar{f}_t$. In this case, as the nonzero attack disturbance probability $p$ approaches zero, attaining exact recovery requires more samples. The controller can counteract this trend by injecting random zero-mean i.i.d. Gaussian inputs into the system following the probabilistic sparsity model with probability $p'$. Under this formulation, $\bar{d}_t$ as the combination of $\bar{e}_t$ and $\bar{f}_t$ continues to satisfy the assumptions required for Theorem 4 and Theorem 6. The nonzero disturbance probability for $\bar{d}_t$ becomes at most $p' + p$, and the exact recovery remains achievable as long as there exist sufficient time periods when both vectors $\bar{e}_t$ and $\bar{f}_t$ are zero, i.e., $p' + p < 1$.

For the bounded basis functions, the sample complexity is minimized when the nonzero disturbance probability $p$ is set to $p^* = 1/3$, as the bound in Theorem 4 scales with $p^{-1}(1-p)^{-2}$. As a result, it is optimal to inject zero-mean i.i.d Gaussian inputs with probability $p' = p^* = 1/3$ as this value minimizes the sample complexity bound. Furthermore, for Lipschitz basis functions, the sample complexity scales as $\max\{(1-p)^{-2}, p^{-1}(1-p)^{-1}\}$ according to Theorem 6. In this case, the optimal $p^*$ value is $p^* = 1/2$, meaning that it is preferable to inject random excitations approximately half of the time when the semi-oblivious attacks are nearly absent, i.e., $p \approx 0$. Finally, although $p$ is unknown to the controller when the probability of attacks is lower than $p^*$ values specified above for bounded and Lipschitz basis functions, injecting excitation signals with zero-mean i.i.d Gaussian inputs at a probability of $p' = p^* - p$ will accelerate the exact recovery rate of the system dynamics.

# 8 Numerical Experiments

We conduct numerical experiments for both the bounded and Lipschitz continuous basis function cases to validate the exact recovery guarantees presented in Sections 5 and 6. Specifically, we examine the convergence behavior of the estimator in (3) under varying values of the sparsity probability $p$ and problem dimensions $(n, m)$. Additionally, we numerically verify the necessary and sufficient conditions established in Section 3. Further numerical experiments exploring additional problem parameters are provided in Appendix C.

**Evaluation metrics.** Given a trajectory $\{x_0, \ldots, x_T\}$, we compute the estimators

$$\hat{A}^{T'} \in \arg\min_{A \in \mathbb{R}^{n \times m}} g_{T'}(A), \quad \forall T' \in \{1, \ldots, T\},$$

where the loss function is defined as $g_{T'}(A) := \sum_{t=0}^{T'-1} \|x_{t+1} - Af(x_t)\|_2$. In our experiments, we solve the convex optimization problem using the CVX solver (Grant & Boyd, 2014). To evaluate the recovery quality of the estimator for each $T'$, we consider the following three metrics:

- The **Loss Gap** is defined as $g_{T'}(\bar{A}) - g_{T'}(\hat{A}_{T'})$. The ground truth $\bar{A}$ is a global solution if and only if the loss gap is 0.

- The **Solution Gap** is defined as $\|\bar{A} - \hat{A}_{T'}\|_F$. The ground truth $\bar{A}$ is the unique solution only if the solution gap is 0.

- The **Optimality Certificate** is defined as

$$\min_{Z \in \mathbb{R}^{n \times m}} \sum_{t \in \mathcal{K}^c} \|Zf(x_t)\|_2 - \sum_{t \in \mathcal{K}} \hat{d}_t^\top Zf(x_t) \quad \text{s.t. } \|Z\|_F \leq 1,$$

which is a convex optimization problem that can be solved using the CVX solver. The ground truth is a global solution if and only if the optimality certificate is 0.

We evaluate these metrics to assess the performance of the estimator in (3) and validate the proposed optimality conditions. For each parameter setting, we independently generate 10 trajectories using the system dynamics in (1) and compute the average of the three metrics.

Since we need to solve estimator (3) many times (for different trajectories and steps $T'$), we consider relatively small-scale problems. In practice, the estimator (3) is only required for $T' = T$, and we only need to solve a single optimization problem. As a result, estimator (3) can be solved for large-scale real-world systems since it is convex and should be solved only once. We provided experiment results for large-scale systems in Appendix C. Next, we explain the experiment setup and the numerical results for the bounded basis functions.

**Bounded basis function.** Given a state space dimension $n$, we set $m = 5n$ and define the basis function as

$$f(x) := \begin{bmatrix} \tilde{f}(x_1) \\ \vdots \\ \tilde{f}(x_n) \end{bmatrix}, \quad \text{where } \tilde{f}(y) := \begin{bmatrix} \sin(y) \\ \vdots \\ \sin(5y) \end{bmatrix}, \quad \forall x \in \mathbb{R}^n, \ y \in \mathbb{R}.$$

The basis function satisfies Assumption 3 with $B = 1$. For each time instance $t \in \mathcal{K}$ and for each $i \in \{1, \ldots, n\}$, the disturbance $\bar{d}_{t,i}$ is independently generated by

$$\bar{d}_{t,i} \sim \text{Uniform}\left(-c_{t,i}\pi, c_{t,i}\pi\right), \quad \text{where } c_{i,t} := \min\{\max\{|x_{t,i}|, 0.1\}, 0.5\}.$$

Here, $\bar{d}_{t,i}$ and $x_{t,i}$ denote $i$-th components of $\bar{d}_t$ and $x_t$, respectively. Since the disturbance distribution is symmetric about the origin, it satisfies Assumption 2. Additionally, as the sine functions $\sin(y), \ldots, \sin(5y)$ are linearly independent, the non-degeneracy condition (Assumption 1) is satisfied. Finally, the ground truth matrix $\bar{A}$ is constructed such that

$$\bar{A}f(x) = \begin{bmatrix} \sum_{k=1}^5 \bar{a}_{1,k} \sin(kx_1) \\ \vdots \\ \sum_{k=1}^5 \bar{a}_{n,k} \sin(kx_n) \end{bmatrix},$$

where the coefficients are sampled independently as

$$\bar{a}_{i,k} \stackrel{\text{i.i.d.}}{\sim} \text{Uniform}(-100, 100), \quad \forall i \in \{1, \ldots, n\}, \ k \in \{1, \ldots, 5\}.$$

We choose a large upper bound for the coefficients $\bar{a}_{i,k}$ (i.e., greater than 1) to illustrate that the stability condition (Assumption 5) is not required in the case of bounded basis functions.

We first compare the performance of estimator (3) under different nonzero disturbance probabilities $p$. Specifically, we set $T = 500$, $n = 1$ and $p \in \{0.7, 0.8, 0.85\}$. The results, presented in Figure 1, align with the Theorem 4. In particular, the optimality certificate accurately reflects the exact recovery of the estimator in (3), and the required sample complexity increases as the attack/disturbance probability $p$ grows.

Next, we analyze the estimator's performance across different problem dimensions $(n, m)$. We set $T = 500$, $p = 0.7$, and evaluate the cases where $n \in \{1, 2, 4\}$. The results are plotted in Figure 2 demonstrate that exact recovery requires more samples as $(n, m)$ increases, further validating the theoretical results established in Theorem 4. After verifying the behavior of the estimator for bounded basis functions, we proceed to simulate its performance using Lipschitz continuous basis functions.

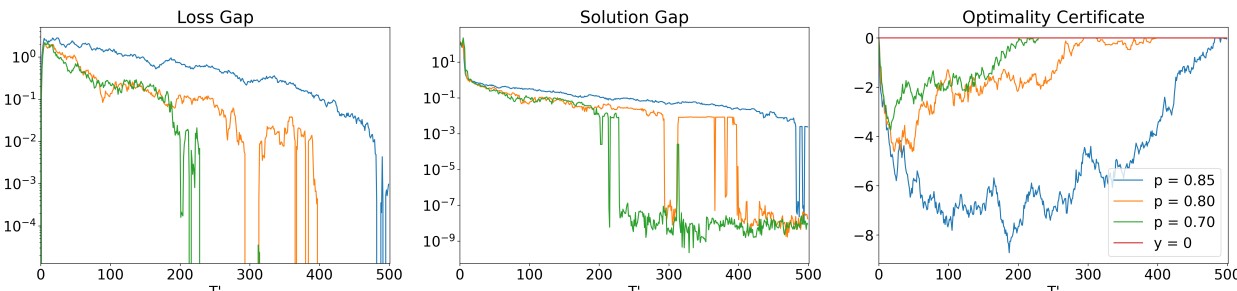

Figure 1: Loss gap, solution gap and optimality certificate of the bounded basis function case with attack/disturbance probability $p = 0.7, 0.8$ and $0.85$.

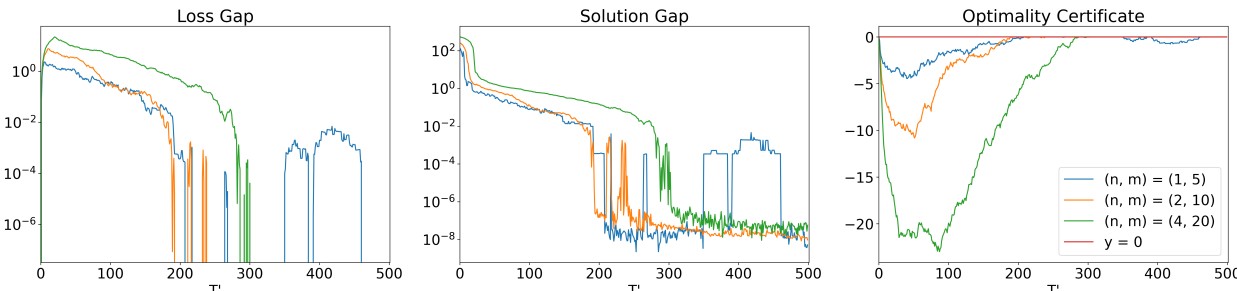

Figure 2: Loss gap, solution gap and optimality certificate of the bounded basis function case with dimension $(n, m) = (1, 5), (2, 10)$ and $(4, 20)$.

**Lipschitz basis function.** Given a state space dimension $n$, we set $m = n$ and define the basis function as

$$f(x) := \frac{1}{\sqrt{n}} \begin{bmatrix} \sqrt{\|x - x_1\|_2^2 + 1} - \sqrt{\|x_1\|_2^2 + 1} \\ \vdots \\ \sqrt{\|x - x_n\|_2^2 + 1} - \sqrt{\|x_n\|_2^2 + 1} \end{bmatrix}, \quad \forall x \in \mathbb{R}^n,$$

where $x_1, \ldots, x_n \in \mathbb{R}^n$ are i.i.d. standard Gaussian random vectors. We can verify that the basis function is Lipschitz continuous with a Lipschitz constant of $L = 1$, thus satisfying Assumption 4. For each time instance $t \in \mathcal{K}$, the disturbance $\bar{d}_t$ is generated by

$$\bar{d}_t := \ell_t \hat{d}_t, \quad \text{where } \ell_t \sim \mathcal{N}(0, \sigma_t^2), \ \hat{d}_t \sim \text{uniform}(\mathbb{S}^{n-1}), \ \ell_t \text{ and } \hat{d}_t \text{ are independent.}$$

Here, we define $\sigma_t^2 := \min\{\|x_t\|_2^2, 1/n\}$. We verify that the random variable $\ell_t$ is zero-mean and sub-Gaussian with parameter $\sigma = 1$. Additionally, since the random vector $\hat{d}_t$ follows a uniform distribution on the unit sphere $\mathbb{S}^{n-1}$, Assumption 6 is satisfied. Note that $\bar{d}_0, \ldots, \bar{d}_{T-1}$ are correlated, violating the i.i.d. assumption commonly made in the literature. Our disturbance model further implies that the intensity of a disturbance vector (represented by $\ell_t$) depends on the current state, which itself is influenced by previous disturbances. Since the points $x_1, \ldots, x_n$ are randomly generated, the multiquadric radial basis functions are linearly independent[1] with probability 1. Thus, the non-degeneracy condition (Assumption 1) is satisfied. Finally, the ground truth matrix $\bar{A}$ is constructed as $U\Sigma V^\top$, where $U, V \in \mathbb{R}^{n \times n}$ are random orthogonal matrices and $\Sigma = \text{diag}(\sigma_1, \ldots, \sigma_n)$ is a diagonal matrix. The singular values $\sigma_i$ are independently drawn from a uniform distribution:

$$\sigma_i \overset{\text{i.i.d.}}{\sim} \text{uniform}(0, \rho), \quad \forall i \in \{1, \ldots, n\},$$

where $\rho > 0$ is the upper bound on the spectral norm of $\bar{A}$.

---

[1] Functions $g_1(y), \ldots, g_k(y)$ are said to be linearly independent if there do not exist constants $c_1, \ldots, c_k$ such that $\sum_{i=1}^{k} c_i g_i(y) = 0$ for all $y$.

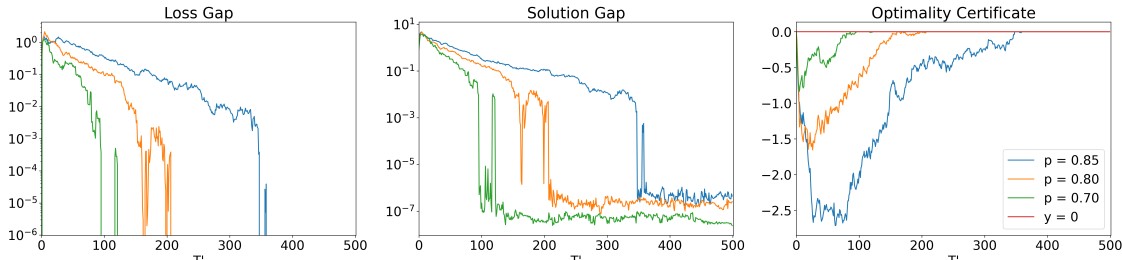

Figure 3: Loss gap, solution gap and optimality certificate of the Lipschitz basis function case with attack/disturbance probability $p = 0.7, 0.8$ and $0.85$.

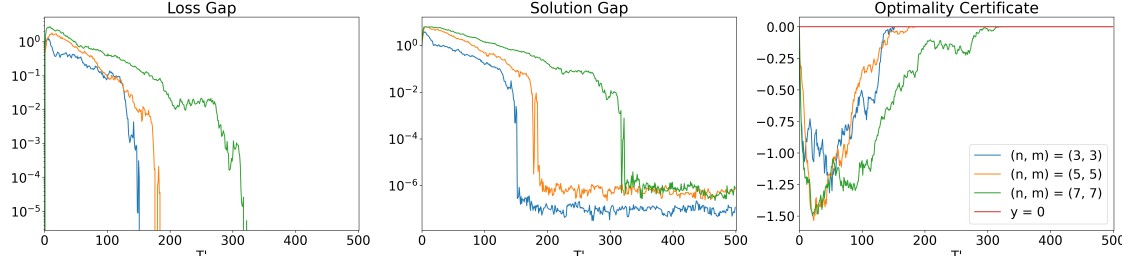

Figure 4: Loss gap, solution gap, and optimality certificate of the Lipschitz basis function case with dimension $(n, m) = (3, 3), (5, 5)$, and $(7, 7)$.

We first evaluate the performance of the estimator in (3) under varying values of the sparsity probability $p$. We set $T = 500$, $n = 3$, and consider $p \in \{0.7, 0.8, 0.85\}$. Additionally, we set the upper bound $\rho$ to be 1, which guarantees the stability condition (Assumption 5). The results are plotted in Figure 3. The results demonstrate that both the loss gap and solution gap converge to zero as the number of samples $T$ increases. This implies that the estimator in (3) achieves exact recovery of the ground truth matrix $\bar{A}$ when a sufficient number of samples is available. Furthermore, the optimality certificate also converges to zero simultaneously with the solution gap, thereby confirming the validity of the necessary and sufficient conditions established in Section 3. Additionally, we observe that the required number of samples increases with the nonzero disturbance probability $p$, which is consistent with the upper bound derived in Theorem 6.

Next, we evaluate the performance of the estimator in (3) across different problem dimensions $(n, m)$. Specifically, we set $T = 500$, $p = 0.75$, and $\rho = 1$, while varying $n \in \{3, 5, 7\}$. The results are presented in Figure 4. We observe that as the problem dimension $(n, m)$ increases, a larger number of samples is required to ensure exact recovery. This finding aligns with the theoretical bound established in Theorem 4.

## 9 Conclusion

This paper addresses the problem of parameterized nonlinear system identification in the presence of *sparse-but-large* disturbances or *semi-oblivious attacks*. The non-smooth estimator (3) is utilized to achieve the exact recovery of the underlying parameter $\bar{A}$. First, we establish necessary and sufficient conditions for the well-specifiedness of the estimator (3) and the uniqueness of optimal solutions to the embedded optimization problem (4). Subsequently, we derive sample complexity bounds for the exact recovery of $\bar{A}$ under two different conditions: bounded basis functions and Lipschitz basis functions. For bounded basis functions, the sample complexity scales as $m^3 n$ in terms of the problem dimension and as $p^{-1}(1 - p)^{-2}$ in terms of the nonzero disturbance probability, up to a logarithm factor. For Lipschitz basis functions, the sample complexity scales as $mn$ in terms of the problem dimension and as $\max\{(1 - p)^{-2}, p^{-1}(1 - p)^{-1}\}$ in terms of the nonzero disturbance probability, up to a logarithm factor. Furthermore, we establish that if the sample complexity scales at an order smaller than $p^{-1}(1 - p)^{-1}$, high-probability exact recovery is unattainable.

This result implies that the term $p^{-1}(1-p)^{-1}$ in our bounds is fundamental and unavoidable. Finally, we validate our theoretical findings through numerical experiments.

**Acknowledgment**

This work was supported by the U.S. Army Research Laboratory and the U.S. Army Research Office under Grant W911NF2010219, the Office of Naval Research under Grant N000142412673, AFOSR, NSF, and the UC Noyce Initiative. Eduardo Sontag was supported by AFOSR FA9550-21-1-0289.

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

# A Comparing Results to Existing Work

**Example 1** (First-order systems). *In the special case when $n = m = 1$ and the basis function is $f(x) = x$, condition (7) reduces to*

$$\left| \sum_{t \in \mathcal{K}} \hat{d}_t x_t \right| \leq \sum_{t \in \mathcal{K}^c} |x_t|,$$

*which is the same as Theorem 1 in Feng & Lavaei (2021).*

**Example 2** (Linear systems). *We consider the case when $m = n$ and the basis function is $f(x) = x$. We also assume the $\Delta$-spaced disturbance model; see the definition in Yalcin et al. (2023). By considering the disturbance period starting at the time step $t_1$, a sufficient condition to guarantee condition (5) is given by*

$$\hat{d}^\top Z \bar{A}^{\Delta-1} \bar{d}_{t_1} \leq \sum_{t=0}^{\Delta-2} \left\| Z \bar{A}^t \bar{d}_{t_1} \right\|_2, \quad \forall Z \in \mathbb{R}^{n \times n}, \tag{13}$$

*where we denote $\hat{d} := \hat{d}_{t_1}$ for simplicity. Let $\hat{D} \in \mathbb{R}^{n \times (n-1)}$ be the matrix of orthonormal bases of the orthogonal complementary space of $f$, namely, $\hat{D}^\top \hat{d} = 0$, $\hat{D}^\top \hat{D} = I_{n-1}$, and $\hat{D}\hat{D}^\top = \boldsymbol{I}_n - \hat{d}\hat{d}^\top$. Then, we can calculate that*

$$\left\| Z \bar{A}^t \bar{d}_{t_1} \right\|_2^2 \geq \left( Z \bar{A}^t \bar{d}_{t_1} \right)^\top \hat{d}\hat{d}^\top \left( Z \bar{A}^t \bar{d}_{t_1} \right),$$

*where the equality holds when $\hat{D}^\top Z \bar{A}^t \bar{d}_{t_1} = 0$, i.e., $Z \bar{A}^t \bar{d}_{t_1}$ is parallel with $\hat{d}$. Therefore, for condition (13) to hold, it is equivalent to consider $Z$ with the form $Z = \hat{d} z^\top$ for some vector $z \in \mathbb{R}^n$. In this case, condition (13) reduces to*

$$z^\top \bar{A}^{\Delta-1} \bar{d}_{t_1} \leq \sum_{t=0}^{\Delta-2} \left| z^\top \bar{A}^t \bar{d}_{t_1} \right|, \quad \forall z \in \mathbb{R}^n. \tag{14}$$

*Condition (14) leads to a better sufficient condition than that in Yalcin et al. (2023). To illustrate the improvement, we consider the special case when the ground truth matrix is $\bar{A} = \lambda \boldsymbol{I}_n$ for some $\lambda \in \mathbb{R}$. Then, condition (14) becomes*

$$|\lambda|^{\Delta-1} \leq \sum_{t=0}^{\Delta-2} |\lambda|^t = \frac{1 - |\lambda|^{\Delta-1}}{1 - |\lambda|} \quad, \text{ which is further equivalent to } |\lambda| + |\lambda|^{1-\Delta} \leq 2,$$

*which is a stronger condition than that in Yalcin et al. (2023). When the disturbance period $\Delta$ is large, we approximately have $|\lambda| \leq 2 - 2^{1-\Delta}$, which is a better condition than that in Figure 1 of Yalcin et al. (2023).*

**Example 3** (First-order linear systems). *In the case when $m = n = 1$ and $f(x) = x$, our results state that the uniqueness of global solutions is equivalent to*

$$\left| \sum_{t \in \mathcal{K}} \hat{d}_t x_t \right| < \sum_{t \in \mathcal{K}^c} |x_t|. \tag{15}$$

*As a comparison, the sufficient condition in Theorem 1 in Feng & Lavaei (2021) is*

$$\sum_{t \in \mathcal{K}} |x_t| < \sum_{t \in \mathcal{K}^c} |x_t|.$$

*Since $|\hat{d}_t| = 1$ for all $t \in \mathcal{K}$, our results (15), as well as Theorem 2, are more general and stronger than that in Feng & Lavaei (2021).*

# B    Proofs

## B.1    Proof of Theorem 1

*Proof of Theorem 1.* Since problem (4) is convex in $A$, the ground truth matrix $\bar{A}$ is a global optimum if and only if

$$0 \in \sum_{t \in \mathcal{K}^c} f(x_t) \otimes \partial \|\mathbf{0}_n\|_2 + \sum_{t \in \mathcal{K}} f(x_t) \otimes \hat{d}_t. \tag{16}$$

See Theorem 4.5 in Clason (2017) for the proof of the above optimality condition. Using the form of the subgradient of the $\ell_2$-norm, condition (16) holds if and only if there exist vectors

$$g_t \in \mathbb{R}^n, \quad \forall t \in \mathcal{K}^c$$

such that

$$\sum_{t \in \mathcal{K}^c} f(x_t) g_t^\top + \sum_{t \in \mathcal{K}} f(x_t) \hat{d}_t^\top = \mathbf{0}_{n \times n}, \quad \|g_t\|_2 \le 1, \quad \forall t \in \mathcal{K}^c. \tag{17}$$

Define the matrices

$$B := \begin{bmatrix} f(x_t) & \forall t \in \mathcal{K}^c \end{bmatrix} \in \mathbb{R}^{m \times (T - |\mathcal{K}|)}, \quad V := \begin{bmatrix} f(x_t) & \forall t \in \mathcal{K} \end{bmatrix} \in \mathbb{R}^{m \times |\mathcal{K}|},$$
$$G := \begin{bmatrix} g_t & \forall t \in \mathcal{K}^c \end{bmatrix} \in \mathbb{R}^{n \times (T - |\mathcal{K}|)}, \quad F := \begin{bmatrix} \hat{d}_t & \forall t \in \mathcal{K} \end{bmatrix} \in \mathbb{R}^{n \times |\mathcal{K}|}.$$

Condition (17) can be written as a combination of second-order cone constraints and linear constraints:

$$\exists G \in \mathbb{R}^{n \times (T - |\mathcal{K}|)}, s, r \in \mathbb{R} \quad \text{s.t. } BG^\top + VF^\top = \mathbf{0}_{m \times n}, \quad \|G_{:,t}\|_2 \le s, \ \forall t, \tag{18}$$
$$s + r = 1, \quad s, r \ge 0,$$

where $G_{:,t}$ is the $t$-th column of $G$ for all $t \in \{1, \ldots, T - |\mathcal{K}|\}$. We define the closed convex cone

$$\mathcal{S} := \left\{ z \in \mathbb{R}^{(T - |\mathcal{K}|)n + 2} \middle| \sqrt{\sum_{i=1}^n z_{(T - |\mathcal{K}|)i + t}^2} \le z_{(T - |\mathcal{K}|)n + 1}, \ \forall t \in \{0, \ldots, T - |\mathcal{K}| - 1\}, \right.$$
$$\left. z_{(T - |\mathcal{K}|)n + 1}, z_{(T - |\mathcal{K}|)n + 2} \ge 0 \right\},$$

and we define the matrix and vector

$$\mathcal{A} := \begin{bmatrix} I_n \otimes B & 0 & 0 \\ 0 & 1 & 1 \end{bmatrix} \in \mathbb{R}^{(mn + 1) \times [(T - |\mathcal{K}|)n + 2]}, \quad b := \begin{bmatrix} -(VF^\top)_{:,1} \\ -(VF^\top)_{:,2} \\ \vdots \\ -(VF^\top)_{:,n} \\ 1 \end{bmatrix} \in \mathbb{R}^{mn + 1},$$

where $(VF^\top)_{:,i}$ is the $i$-th column of $VF^\top$. Then, condition (18) can be equivalently written as

$$\exists z \in \mathbb{R}^{(T - |\mathcal{K}|)n + 2} \quad \text{s.t. } \mathcal{A}z = b, \quad z \in \mathcal{S}. \tag{19}$$

Since the cone $\mathcal{S}$ is closed and convex, we can apply the *generalized Farka's lemma* to conclude that condition (19) is equivalent to

$$\forall y \in \mathbb{R}^{mn + 1}, \quad \left( \mathcal{A}^\top y \in \mathcal{S}^* \implies b^\top y \ge 0 \right), \tag{20}$$

where $\mathcal{S}^*$ is the dual cone of $\mathcal{S}$. It can be verified that the dual cone is

$$\mathcal{S}^* = \left\{ z \in \mathbb{R}^{(T-|\mathcal{K}|)n+2} \,\middle|\, \sum_{t=0}^{T-|\mathcal{K}|-1} \sqrt{\sum_{i=1}^{n} z_{(T-|\mathcal{K}|)i+t}^2} \leq z_{(T-|\mathcal{K}|)n+1}, z_{(T-|\mathcal{K}|)n+1}, z_{(T-|\mathcal{K}|)n+2} \geq 0 \right\}.$$

We can equivalently write condition (20) as

$$\forall Z \in \mathbb{R}^{n \times m}, \ p \in \mathbb{R}, \quad \left( \|ZB\|_{2,1} \leq p, \quad p \geq 0 \implies \langle VF^\top, Z^\top \rangle \leq p \right),$$

By eliminating variable $p$, we get

$$\langle VF^\top, Z^\top \rangle \leq \|ZB\|_{2,1}, \quad \forall Z \in \mathbb{R}^{n \times m},$$

where the $\ell_{2,1}$-norm is defined as

$$\|M\|_{2,1} := \sum_{j=1}^{n} \sqrt{\sum_{i=1}^{m} M_{ij}^2}, \quad \forall M \in \mathbb{R}^{n \times m}.$$

The above condition is equivalent to condition (5), and this completes the proof. $\qquad\square$

## B.2 Proof of Corollary 2

*Proof of Corollary 2.* We choose

$$Z := \frac{\sum_{t \in \mathcal{K}} \hat{d}_t f(x_t)^\top}{\left\| \sum_{t \in \mathcal{K}} \hat{d}_t f(x_t)^\top \right\|_F}.$$

Then, condition (5) implies

$$\left\| \sum_{t \in \mathcal{K}} f(x_t) \hat{d}_t^\top \right\|_F = \sum_{t \in \mathcal{K}} \hat{d}_t^\top Z f(x_t) \leq \sum_{t \in \mathcal{K}^c} \|Z f(x_t)\|_2 \leq \sum_{t \in \mathcal{K}^c} \|f(x_t)\|_2,$$

where the last step is because $\|Z\|_2 \leq \|Z\|_F = 1$. Now, suppose that the basis dimension is $m = 1$. In this case, we have

$$\sum_{t \in \mathcal{K}} \hat{d}_t^\top Z f(x_t) = \left( \sum_{t \in \mathcal{K}} f(x_t) \hat{d}_t \right)^\top Z^\top \leq \left\| \sum_{t \in \mathcal{K}} f(x_t) \hat{d}_t \right\|_F \|Z\|_2,$$

$$\sum_{t \in \mathcal{K}^c} \|Z f(x_t)\|_2 = \sum_{t \in \mathcal{K}^c} |f(x_t)| \|Z\|_2 = \sum_{t \in \mathcal{K}^c} \|f(x_t)\|_2 \|Z\|_2.$$

Combining the above two inequalities shows that condition (7) is also a sufficient condition. $\qquad\square$

## B.3 Proof of Theorem 2

We establish the sufficient and the necessary parts of Theorem 2 by the following two lemmas.

**Lemma 1** (Sufficient condition for uniqueness). *Suppose that condition (5) holds. If for every nonzero $Z \in \mathbb{R}^{n \times m}$ such that*

$$\sum_{t \in \mathcal{K}} \hat{d}_t^\top Z f(x_t) = \sum_{t \in \mathcal{K}^c} \|Z f(x_t)\|_2,$$

*it holds that*

$$\sum_{t \in \mathcal{K}} \left| \hat{d}_t^\top Z f(x_t) \right| < \sum_{t \in \mathcal{K}} \|Z f(x_t)\|_2.$$

*Then, the ground truth matrix $\bar{A}$ is the unique global solution to problem (4).*

*Proof.* The ground truth $\bar{A}$ is the unique solution if and only if for every matrix $A \in \mathbb{R}^{n \times m}$ such that $A \neq \bar{A}$, the loss function of $A$ is larger than that of $\bar{A}$, namely,

$$\sum_{t \in \mathcal{K}} \|\bar{d}_t\|_2 < \sum_{t \in \mathcal{K}^c} \|(\bar{A} - A)f(x_t)\|_2 + \sum_{t \in \mathcal{K}} \|(\bar{A} - A)f(x_t) + \bar{d}_t\|_2. \tag{21}$$

Denote

$$Z := A - \bar{A} \in \mathbb{R}^{n \times m}.$$

The inequality (21) becomes

$$\sum_{t \in \mathcal{K}^c} \| - Zf(x_t)\|_2 + \sum_{t \in \mathcal{K}} \left( \| - Zf(x_t) + \bar{d}_t\|_2 - \|\bar{d}_t\|_2 \right) > 0. \tag{22}$$

Since problem (4) is convex in $A$, it is sufficient to guarantee that $\bar{A}$ is a strict local minimum. Therefore, the uniqueness of global solutions can be formulated as

$$\text{condition (22) holds}, \quad \forall Z \in \mathbb{R}^{n \times m} \quad \text{s.t. } 0 < \|Z\|_F \leq \epsilon, \tag{23}$$

where $\epsilon > 0$ is a sufficiently small constant. In the following, we fix the direction $Z$ and discuss two different cases.

**Case I.** We first consider the case when condition (5) holds strictly, namely,

$$\sum_{t \in \mathcal{K}^c} \|Zf(x_t)\|_2 - \sum_{t \in \mathcal{K}} \hat{d}_t^\top Zf(x_t) > 0.$$

Since the $\ell_2$-norm is a convex function, it holds that

$$\| - Zf(x_t) + \bar{d}_t\|_2 - \|\bar{d}_t\|_2 \geq \left\langle \partial \|\bar{d}_t\|_2, -Zf(x_t) \right\rangle = -\hat{d}_t^\top Zf(x_t).$$

Therefore, we get

$$\sum_{t \in \mathcal{K}^c} \| - Zf(x_t)\|_2 + \sum_{t \in \mathcal{K}} \left( \| - Zf(x_t) + \bar{d}_t\|_2 - \|\bar{d}_t\|_2 \right) \geq \sum_{t \in \mathcal{K}^c} \| - Zf(x_t)\|_2 + \sum_{t \in \mathcal{K}} -\hat{d}_t^\top Zf(x_t) > 0,$$

which exactly leads to inequality (22).

**Case II.** Next, we consider the case when

$$\sum_{t \in \mathcal{K}} \hat{d}_t^\top Zf(x_t) = \sum_{t \in \mathcal{K}^c} \|Zf(x_t)\|_2, \quad \sum_{t \in \mathcal{K}} \left| \hat{d}_t^\top Zf(x_t) \right| < \sum_{t \in \mathcal{K}} \|Zf(x_t)\|_2. \tag{24}$$

Since $\epsilon$ is a sufficiently small constant, we know

$$\bar{d}_t^\alpha := -\alpha Zf(x_t) + \bar{d}_t \neq 0, \quad \forall \alpha \in [0, 1],$$

and the $\ell_2$-norm is second-order continuously differentiable in an open set that contains the line. Therefore, the *mean value theorem* implies that there exists $\alpha \in [0, 1]$ such that for each $t \in \mathcal{K}$, it holds

$$\| - Zf(x_t) + \bar{d}_t\|_2 - \|\bar{d}_t\|_2 = \left\langle \hat{d}_t, -Zf(x_t) \right\rangle + \frac{1}{2} [-Zf(x_t)]^\top \left( \frac{\boldsymbol{I}_n}{\|\bar{d}_t^\alpha\|_2} - \frac{\bar{d}_t^\alpha \left(\bar{d}_t^\alpha\right)^\top}{\|\bar{d}_t^\alpha\|_2^3} \right) [-Zf(x_t)]. \tag{25}$$

We can calculate that

$$[-Zf(x_t)]^\top \left( \frac{I}{\|\bar{d}_t^\alpha\|_2} - \frac{\bar{d}_t^\alpha \left(\bar{d}_t^\alpha\right)^\top}{\|\bar{d}_t^\alpha\|_2^3} \right) [-Zf(x_t)] = \frac{\|Zf(x_t)\|_2^2}{\|\bar{d}_t^\alpha\|_2} - \frac{\left\langle \bar{d}_t^\alpha, Zf(x_t) \right\rangle^2}{\|\bar{d}_t^\alpha\|_2^3} \geq 0, \tag{26}$$

where the equality holds if and only if $Zf(x_t)$ is parallel with $\bar{d}_t^\alpha$. By the definition of $\bar{d}_t^\alpha$, the equality holds if and only if $Zf(x_t)$ is parallel with $\bar{d}_t$, which is further equivalent to

$$\left| \left\langle \hat{d}_t, Zf(x_t) \right\rangle \right| = \|Zf(x_t)\|_2.$$

Substituting (25) and (26) into (22), we have

$$\sum_{t \in \mathcal{K}^c} \| - Zf(x_t)\|_2 + \sum_{t \in \mathcal{K}} \left( \| - Zf(x_t) + \bar{d}_t\|_2 - \|\bar{d}_t\|_2 \right) \geq \sum_{t \in \mathcal{K}^c} \|Zf(x_t)\|_2 - \sum_{t \in \mathcal{K}} \left\langle \hat{d}_t, Zf(x_t) \right\rangle = 0,$$

where the equality holds if and only if

$$\left| \left\langle \hat{d}_t, Zf(x_t) \right\rangle \right| = \|Zf(x_t)\|_2, \quad \forall t \in \mathcal{K}.$$

Considering the second condition in (24), the above equality condition is violated by some $t \in \mathcal{K}$. Therefore, we have proven that condition (22) holds strictly.

Combining the two cases, we complete the proof. $\qquad\square$

Next, we prove that the condition in Lemma 1 is also necessary for the uniqueness.

**Lemma 2** (Necessary condition for uniqueness). *Suppose that condition (5) holds. If the ground truth matrix $\bar{A}$ is the unique global solution to problem (4), then for every nonzero $Z \in \mathbb{R}^{n \times m}$, we have*

$$\sum_{t \in \mathcal{K}} \hat{d}_t^\top Zf(x_t) < \sum_{t \in \mathcal{K}^c} \|Zf(x_t)\|_2 \quad or \quad \sum_{t \in \mathcal{K}} \left| \hat{d}_t^\top Zf(x_t) \right| < \sum_{t \in \mathcal{K}} \|Zf(x_t)\|_2. \tag{27}$$

*Proof.* Assume conversely that there exists a nonzero $Z \in \mathbb{R}^{n \times m}$ such that

$$\sum_{t \in \mathcal{K}} \hat{d}_t^\top Zf(x_t) = \sum_{t \in \mathcal{K}^c} \|Zf(x_t)\|_2, \quad \sum_{t \in \mathcal{K}} \left| \hat{d}_t^\top Zf(x_t) \right| = \sum_{t \in \mathcal{K}} \|Zf(x_t)\|_2. \tag{28}$$

Without loss of generality, we assume that

$$0 < \|Z\|_2 \leq \epsilon$$

for a sufficiently small $\epsilon$. In this case, the second condition in (28) implies that

$$\left| \hat{d}_t^\top Zf(x_t) \right| = \|Zf(x_t)\|_2, \text{ and } \quad Zf(x_t) \text{ is parallel with } \bar{d}_t, \quad \forall t \in \mathcal{K}.$$

Therefore, when $\epsilon$ is sufficiently small, equations (26) and (24) lead to

$$\| - Zf(x_t) + \bar{d}_t\|_2 - \|\bar{d}_t\|_2 = - \left\langle \hat{d}_t, Zf(x_t) \right\rangle, \quad \forall t \in \mathcal{K}.$$

We now show that condition (22) fails:

$$\sum_{t \in \mathcal{K}^c} \| - Zf(x_t)\|_2 + \sum_{t \in \mathcal{K}} \left( \| - Zf(x_t) + \bar{d}_t\|_2 - \|\bar{d}_t\|_2 \right) = \sum_{t \in \mathcal{K}} \left\langle \hat{d}_t, Zf(x_t) \right\rangle - \sum_{t \in \mathcal{K}} \left\langle \hat{d}_t, Zf(x_t) \right\rangle = 0.$$

This contradicts the assumption that $\bar{A}$ is the unique solution to the problem (4). $\qquad\square$

Combining Lemmas 1 and 2, we have the following necessary and sufficient condition for the uniqueness of the ground truth solution $\bar{A}$.

### B.4 Proof of Theorem 3

*Proof.* Without loss of generality, we can assume that $\bar{A}$ is full-rank. Otherwise, the system space can be reduced to the subspace spanned by $\bar{A}$. Similarly, we assume that $\bar{A}$ is diagonalizable, such that $\bar{A} = P\bar{\Lambda}P^{-1}$. If this is not the case, define $\tilde{A} = \bar{A} + \epsilon \boldsymbol{I}_n$ for a sufficiently small $\epsilon$ to ensure diagonalizability. Taking the limit as $\epsilon \to 0$ preserves the validity of the argument.

We redefine the linear system updates $x_{t+1} = \bar{A}x_t + \bar{d}_t$ as follows:

$$x_{t+1} = \bar{A}x_t + \bar{d}_t \Rightarrow \tilde{x}_{t+1} = \bar{\Lambda}\tilde{x}_t + \tilde{d}_t,$$

where $\tilde{x}_t = P^{-1}x_t$ and $\tilde{d}_t = P^{-1}\bar{d}_t$. This transformation can be thought of as coordinate transformations in $n$ dimensional space. Thus, we can establish the result for the LTI systems with a diagonal ground truth matrix $\bar{\Lambda}$, whose diagonal entries are $\bar{\lambda}_1, \ldots, \bar{\lambda}_n$. In this case, the system update equations decompose into $n$ separate equations of the form: $\tilde{x}_{t+1}^i = \bar{\lambda}_i \tilde{x}_t^i + \tilde{d}_t^i, \forall i = 1, \ldots, n$ were $\tilde{x}_t^i \in \mathbb{R}$ is the i-th element of the system state at time $t$ in the new coordinate system.

Now, suppose that the nonzero disturbance vectors are chosen from the subspace $\tilde{D} := \{\tilde{d} : P\tilde{d} \in D\}$ where $D = \{\bar{d} : \tilde{d} = P^{-1}\bar{d}, \tilde{d}^1 = 0\}$. Due to the constraint on the first entry of the $\tilde{d}$, $\tilde{d}^1$, the subspace $D$ has dimension less than $n$. Consequently, since $P^{-1}x_0^1 = \tilde{x}_0^1 = 0$, it follows that $\tilde{x}_t^1 = 0, \forall t \geq 0$. Note that setting the first entry to zero is done without loss of generality, as this argument applies to any coordinate or any subset of coordinates.

Next, define $\hat{\Lambda}$ as the diagonal matrix with diagonal entries $(\hat{\lambda}_1, \bar{\lambda}_2, \ldots, \bar{\lambda}_n)$ where $\hat{\lambda}_1 \neq \bar{\lambda}_1$, where all the diagonal entries are identical to those of $\bar{\Lambda}$ except for the first one. Suppose that we start two dynamical systems with ground truth matrices $\bar{A} = P\bar{\Lambda}P^{-1}$ and $\hat{A} = P\hat{\Lambda}P^{-1}$, where the nonzero disturbances $\{\bar{d}_t\}_{t=0}^{T-1}$ are chosen from $D$. Since $\tilde{x}_t^1 = 0, \forall t \geq 0$ in both systems, their trajectories over time must be identical, i.e., $\{\tilde{x}_t\}_{t=0}^T$ is the same for both systems.

Suppose, for contradiction, that there exists an estimator that uniquely recovers $\bar{\Lambda}$ in finite time $T$, given by:

$$\bar{\Lambda} \in \arg\min_{\Lambda \in \text{Diag}(n)} g(\Lambda; \{\tilde{x}_t\}_{t=0}^T),$$

where the decision variable $\Lambda$ is restricted to be a diagonal matrix. Since $\bar{\Lambda}$ is the unique solution, we must have

$$g(\bar{\Lambda}; \{\tilde{x}_t\}_{t=0}^T) < g(\hat{\Lambda}; \{\tilde{x}_t\}_{t=0}^T)$$

However, since the objective function $g(\Lambda; \{\tilde{x}_t\}_{t=0}^T)$ depends solely on the system state over time, and these states are identical for both systems, it follows that $g(\hat{\Lambda}; \{\tilde{x}_t\}_{t=0}^T) = g(\bar{\Lambda}; \{\tilde{x}_t\}_{t=0}^T)$. This contradicts the uniqueness assumption of the solution, which concludes the proof. $\square$

### B.5 Proof of Theorem 4

*Proof of Theorem 4.* Since both sides of inequality (9) are affine in $Z$, it suffices to prove that

$$\mathbb{P}\left[\hat{d}_1(Z) - \hat{d}_2(Z) < 0, \ \forall Z \in \mathbb{S}_F\right] \geq 1 - \delta, \tag{29}$$

where $\mathbb{S}_F$ is the Frobenius-norm unit sphere in $\mathbb{R}^{n \times m}$ and

$$\hat{d}_1(Z) := \sum_{t \in \mathcal{K}} \langle Z^\top, f(x_t)\hat{d}_t^\top \rangle, \quad \hat{d}_2(Z) := \sum_{t \in \mathcal{K}^c} \|Zf(x_t)\|_2.$$

The proof is divided into two steps.

**Step 1.** First, we fix the vector $Z \in \mathbb{S}_F$ and prove that

$$\mathbb{P}\left[\hat{d}_1(Z) - \hat{d}_2(Z) < -\theta\right] \geq 1 - \delta,$$

holds for some constant $\theta > 0$. Using Markov's inequality, it is sufficient to prove that for some $\nu > 0$, it holds that

$$\mathbb{E}\left[\exp\left(\nu\left[\hat{d}_1(Z) - \hat{d}_2(Z)\right]\right)\right] \leq \exp(-\nu\theta)\delta. \tag{30}$$

We focus on the case when $\mathcal{K}$ is not empty, which happens with high probability. The proof of this step is also divided into two sub-steps.

**Step 1-1.** We first analyze the term $\hat{d}_1(Z)$. Let $T'$ be the last nonzero disturbance time instance, i.e.,

$$T' := \max\{t \mid t \in \mathcal{K}\}.$$

Then, we have

$$\mathbb{E}\left[\exp\left[\nu\hat{d}_1(Z)\right]\right] = \mathbb{E}\left[\exp\left(\nu\sum_{t\in\mathcal{K}\backslash\{T'\}}\left\langle Z^\top, f(x_t)\hat{d}_t^\top\right\rangle\right) \times \mathbb{E}\left[\exp\left[\nu\left\langle Z^\top, f(x_{T'})\hat{d}_{T'}^\top\right\rangle\right] \mid \mathcal{F}_{T'}\right]\right]. \tag{31}$$

According to Assumption 2, the direction $\hat{d}_{T'}$ is a unit vector. Since

$$\left|[Zf(x_{T'})]^\top \hat{d}_{T'}\right| \leq \|Zf(x_{T'})\|_2 \leq \|Z\|_2\|f(x_{T'})\|_2 \leq \|Z\|_F\sqrt{m}\|f(x_{T'})\|_\infty \leq \sqrt{m}B,$$

the random variable $[Zf(x_{T'})]^\top \hat{d}_{T'}$ is sub-Gaussian with parameter $mB^2$. Therefore, the property of sub-Gaussian random variables implies that

$$\mathbb{E}\left[\exp\left[\nu\left\langle Z^\top, f(x_{T'})\hat{d}_{T'}^\top\right\rangle\right] \mid \mathcal{F}_{T'}\right] \leq \exp\left(\frac{\nu^2 \cdot mB^2}{2}\right).$$

Substituting into (31), we get

$$\mathbb{E}\left[\exp\left[\nu\hat{d}_1(Z)\right]\right] \leq \mathbb{E}\left[\exp\left(\nu\sum_{t\in\mathcal{K}\backslash\{T'\}}\left\langle Z^\top, f(x_t)\hat{d}_t^\top\right\rangle\right)\right] \cdot \exp\left(\frac{\nu^2 \cdot mB^2}{2}\right).$$

Continuing this process for all $t \in \mathcal{K}$, it follows that

$$\mathbb{E}\left[\exp\left[\nu\hat{d}_1(Z)\right]\right] \leq \exp\left(\frac{\nu^2 \cdot mB^2|\mathcal{K}|}{2}\right). \tag{32}$$

**Step 1-2.** Now, we consider the second term in (30), namely, $-\hat{d}_2(Z)$. Define

$$\mathcal{K}' := \{t \mid 1 \leq t \leq T, \ t \in \mathcal{K}^c, \ t-1 \in \mathcal{K}\}.$$

With probability at least $1 - \exp[-\Theta[p(1-p)T]]$, we have

$$|\mathcal{K}'| = \Theta[p(1-p)T].$$

Therefore, $\mathcal{K}'$ is non-empty with high probability. Since $\|Zf(x_t)\|_2 \geq 0$ for all $t \in \mathcal{K}^c$, we have

$$\mathbb{E}\left[\exp\left[-\nu\hat{d}_2(Z)\right]\right] \leq \mathbb{E}\left[\exp\left(-\nu\sum_{t\in\mathcal{K}'}\|Zf(x_t)\|_2\right)\right] \tag{33}$$

$$= \mathbb{E}\left[\exp\left(-\nu\sum_{t\in\mathcal{K}'\backslash\{T'\}}\|Zf(x_t)\|_2\right) \times \mathbb{E}\left[\exp\left(-\nu\|Zf(x_{T'})\|_2\right) \mid \mathcal{F}_{T'}\right]\right],$$

where $T'$ is the last time instance in $\mathcal{K}'$, namely,

$$T' := \max\{t \mid t \in \mathcal{K}'\}.$$

By Bernstein's inequality (Wainwright, 2019), we can estimate that

$$\mathbb{E}\left[\exp\left(-\nu\|Zf(x_{T'})\|_2\right) \mid \mathcal{F}_{T'}\right] \leq \exp\left[-\nu\mathbb{E}\left(\|Zf(x_{T'})\|_2 \mid \mathcal{F}_{T'}\right) + \frac{\nu^2}{2}\mathbb{E}\left(\|Zf(x_{T'})\|_2^2 \mid \mathcal{F}_{T'}\right)\right]$$

$$\leq \exp\left[-\frac{\nu}{\sqrt{m}B}\mathbb{E}\left(\|Zf(x_{T'})\|_2^2 \mid \mathcal{F}_{T'}\right) + \frac{\nu^2}{2}\mathbb{E}\left(\|Zf(x_{T'})\|_2^2 \mid \mathcal{F}_{T'}\right)\right],$$

where the last inequality is from

$$\|Zf(x_{T'})\|_2 \leq \sqrt{m}B.$$

Assumption 1 implies that

$$\mathbb{E}\left(\|Zf(x_{T'})\|_2^2 \mid \mathcal{F}_{T'}\right) = \left\langle ZZ^\top, \mathbb{E}\left[f(x_{T'})f(x_{T'})^\top \mid \mathcal{F}_{T'}\right]\right\rangle \geq \lambda^2\|Z\|_F^2 = \lambda^2.$$

If we choose $\nu$ such that

$$0 < \nu < \frac{2}{\sqrt{m}B}, \tag{34}$$

we have

$$\mathbb{E}\left[\exp\left(-\nu\|Zf(x_{T'})\|_2\right) \mid \mathcal{F}_{T'}\right] \leq \exp\left[\left(\frac{\nu^2}{2} - \frac{\nu}{\sqrt{m}B}\right)\lambda^2\right].$$

Substituting into inequality (33), it follows that

$$\mathbb{E}\left[\exp\left[-\nu\hat{d}_2(Z)\right]\right] \leq \mathbb{E}\left[\exp\left(-\nu\sum_{t\in\mathcal{K}'\backslash\{T'\}}\|Zf(x_t)\|_2\right) \times \exp\left[\left(\frac{\nu^2}{2} - \frac{\nu}{\sqrt{m}B}\right)\lambda^2\right]\right].$$

Continuing this process for all $t \in \mathcal{K}'$, we have

$$\mathbb{E}\left[\exp\left[-\nu\hat{d}_2(Z)\right]\right] \leq \exp\left[\left(\frac{\nu^2}{2} - \frac{\nu}{\sqrt{m}B}\right)\lambda^2|\mathcal{K}'|\right]. \tag{35}$$

Combining the inequalities (32) and (35), we have

$$\mathbb{E}\left[\exp\left(\nu\left[\hat{d}_1(Z) - \hat{d}_2(Z)\right]\right)\right] \leq \exp\left[\frac{m\nu^2B^2}{2}|\mathcal{K}| + \left(\frac{\nu^2}{2} - \frac{\nu}{\sqrt{m}B}\right)\lambda^2|\mathcal{K}'|\right].$$

We choose

$$\theta := \frac{\lambda^2p(1-p)T}{4\sqrt{m}B}.$$

In order to satisfy condition (30), it is equivalent to have

$$\frac{m\nu^2B^2}{2}|\mathcal{K}| + \left(\frac{\nu^2}{2} - \frac{\nu}{\sqrt{m}B}\right)\lambda^2|\mathcal{K}'| + \frac{\lambda^2\nu p(1-p)T}{4\sqrt{m}B} \leq \log(\delta). \tag{36}$$

Now, we consider the fact that $\mathcal{K}$ is generated by the probabilistic sparsity model. Using the Bernoulli bound, it holds with probability at least $1 - \exp[-\Theta[p(1-p)T]]$ that

$$|\mathcal{K}| \leq 2pT, \quad |\mathcal{K}'| \geq \frac{p(1-p)T}{2}. \tag{37}$$

Thus, with the same probability, we have the estimation

$$\frac{m\nu^2 B^2}{2}|\mathcal{K}| + \left(\frac{\nu^2}{2} - \frac{\nu}{\sqrt{m}B}\right)\lambda^2|\mathcal{K}'| + \frac{\lambda^2\nu p(1-p)T}{4\sqrt{m}B} \le \frac{m\nu^2 B^2}{2} \cdot 2pT + \left(\frac{\nu^2}{2} - \frac{\nu}{2\sqrt{m}B}\right)\lambda^2 \cdot \frac{p(1-p)T}{2}.$$

Note that the inequality holds for the term containing $\mathcal{K}'$ since the coefficient is negative under condition (34). Choosing

$$\nu := \frac{\lambda^2(1-p)}{2\sqrt{m}B[4mB^2 + \lambda^2(1-p)]},$$

we get

$$\frac{m\nu^2 B^2}{2}|\mathcal{K}| + \left(\frac{\nu^2}{2} - \frac{\nu}{\sqrt{m}B}\right)\lambda^2|\mathcal{K}'| + \frac{\lambda^2\nu p(1-p)T}{4\sqrt{m}B} \le -\frac{p(1-p)^2}{16m\kappa^2(4m\kappa^2 + 1 - p)} \cdot T,$$

where we define $\kappa := B/\lambda \ge 1$. Note that our choice of $\nu$ satisfies the condition (34). Therefore, in order for inequality (36) to hold, the sample complexity should satisfy

$$T \ge \frac{16m\kappa^2(4m\kappa^2 + 1 - p)}{p(1-p)^2} \log\left(\frac{1}{\delta}\right).$$

By considering the Bernoulli bound (37), the sample complexity bound becomes

$$T = \Omega\left[\max\left\{\frac{m\kappa^2(m\kappa^2 + 1 - p)}{p(1-p)^2}, \frac{1}{p(1-p)}\right\}\log\left(\frac{1}{\delta}\right)\right] \tag{38}$$

$$= \Omega\left[\frac{m^2\kappa^4}{p(1-p)^2}\log\left(\frac{1}{\delta}\right)\right].$$

**Step 2.** Next, we establish the bound (29) by discretization techniques. More specifically, suppose that $\epsilon > 0$ is a constant and $\{Z^1, \dots, Z^N\} \subset \mathbb{S}_F$ is an $\epsilon$-net of the sphere $\mathbb{S}_F$ under the Frobenius norm, where we can bound

$$\log(N) \le mn \cdot \log\left(1 + \frac{2}{\epsilon}\right).$$

Then, for every $Z \in \mathbb{S}_F$, we can find a point in the $\epsilon$-net, denoted as $Z'$, such that

$$\|Z - Z'\|_F \le \epsilon.$$

Now, we upper bound the difference $f(Z) - f(Z')$, where we define the function

$$f(Z) := \hat{d}_1(Z) - \hat{d}_2(Z), \quad \forall Z \in \mathbb{R}^{n \times m}.$$

We can calculate that

$$\begin{aligned}
f(Z) - f(Z') &= \sum_{t \in \mathcal{K}} \hat{d}_t(Z - Z')f(x_t) - \sum_{t \in \mathcal{K}^c}(\|Zf(x_t)\|_2 - \|Z'f(x_t)\|_2) \\
&\le \sum_{t \in \mathcal{K}} \hat{d}_t(Z - Z')f(x_t) + \sum_{t \in \mathcal{K}^c}\|(Z - Z')f(x_t)\|_2 \\
&\le \sum_{t \in \mathcal{K}}\|Z - Z'\|_F\|f(x_t)\hat{d}_t^\top\|_F + \sum_{t \in \mathcal{K}^c}\|Z - Z'\|_2\|f(x_t)\|_2 \\
&\le \sum_{t \in \mathcal{K}}\|Z - Z'\|_F\|f(x_t)\|_2 + \sum_{t \in \mathcal{K}^c}\|Z - Z'\|_F\|f(x_t)\|_2 \\
&\le T \cdot \epsilon\sqrt{m}B = \sqrt{m}TB \cdot \epsilon.
\end{aligned}$$

We choose

$$\epsilon := \frac{\theta}{\sqrt{m}TB} = \Theta\left[\frac{p(1-p)}{m\kappa^2}\right].$$

Therefore, under the event that

$$f(Z^i) < -\theta, \quad \forall i = 1, \ldots, N, \tag{39}$$

we have

$$f(Z) < -\theta + \sqrt{m}TB \cdot \epsilon = 0, \quad \forall Z \in \mathbb{S}_F.$$

Hence, it suffices to estimate the probability that event (39) happens. To bound the failing probability, we replace $\delta$ with $\delta/N$ in (38) and it follows that

$$\mathbb{P}\left[f(Z^i) < -\theta\right] \geq 1 - \frac{\delta}{N}, \quad \forall i = 1, \ldots, N.$$

Applying the union bound over all $i \in \{1, \ldots, N\}$, the event (39) happens with probability at least $1 - \delta$, namely,

$$\mathbb{P}\left[f(Z^i) < -\theta, \ \forall i = 1, \ldots, N\right] \geq 1 - \delta.$$

With this choice of $\delta$, the sample complexity should be at least

$$T = \Omega\left[\frac{m^2\kappa^4}{p(1-p)^2}\log\left(\frac{N}{\delta}\right)\right] = \Omega\left[\frac{m^2\kappa^4}{p(1-p)^2}\left[mn\log\left(\frac{m\kappa}{p(1-p)}\right) + \log\left(\frac{1}{\delta}\right)\right]\right].$$

This completes the proof. $\qquad\square$

### B.6 Proof of Theorem 5

*Proof of Theorem 5.* We only need to show that condition (8) fails with probability at least $1 - \exp(-m/3)$. We choose the matrix

$$\bar{A} := \begin{bmatrix} 1 & 0_{1\times(m-1)} \\ 0_{n-1} & 0_{(n-1)\times(m-1)} \end{bmatrix} \in \mathbb{R}^{n\times m}.$$

As a result, the last $n-1$ elements of $\bar{A}f(x)$ are zero for every state $x \in \mathbb{R}^n$. Moreover, we will choose the basis function $f$ such that its values will only depend on the first element of state $x \in \mathbb{R}^n$. With these definitions, the dynamics of $x_t$ reduces to the dynamics of its first element $(x_t)_1$. Hence, we can assume without loss of generality that $n = 1$ in the remainder of the proof.

We define the basis function $f : \mathbb{R} \mapsto \mathbb{R}^m$ as

$$\tilde{f}(x) := \left[\frac{x}{\max\{|x|,1\}} \quad \sin(x) \quad \sin(2x) \quad \cdots \quad \sin[(m-1)x]\right], \quad \forall x \in \mathbb{R}.$$

Under the above definitions, it is straightforward to show that the following properties hold, and we omit the proof:

$$f(0) = \mathbf{0}_m, \quad f\left[\bar{A}f(x)\right] = f(x), \quad \forall x \in \mathbb{R}. \tag{40}$$

Finally, the attack/disturbance vector is defined as

$$\bar{d}_t|\mathcal{F}_t \sim \text{Uniform}\left\{[-(|x_t| + 2\pi), -(|x_t| + \pi)] \cup [|x_t| + \pi, |x_t| + 2\pi]\right\}, \quad \forall t \in \mathcal{K}.$$

The remainder of the proof is divided into three steps.

**Step 1.** In the first step, we prove that Assumptions 2-1 hold. By the definition of $f(x)$, we have

$$\|f(x)\|_\infty = \max\left\{\frac{|x|}{\max\{|x|, 1\}}, |\sin(x)|, \ldots, |\sin[(m-1)x]|\right\} \leq 1, \quad \forall x \in \mathbb{R},$$

which implies that Assumption 3 holds with $B = 1$. Moreover, the semi-oblivious condition (Assumption 2) is a result of the symmetric distribution of $\bar{d}_t|\mathcal{F}_t$.

Finally, we prove that Assumption 1 holds. For the notational simplicity, in this step, we omit the subscript $t$, the conditioning on the filtration $\mathcal{F}_t$, and the event $t \in \mathcal{K}$. The model of disturbance vector $d$ implies that

$$|x + d| \geq |d| - |x| \geq \pi > 1.$$

Therefore, we have

$$f(x + d) = \begin{bmatrix} \frac{x+d}{|x+d|} & \sin[(x+d)] & \cdots & \sin[(m-1)(x+d)] \end{bmatrix}.$$

For any vector $\nu \in \mathbb{R}^m$, we want to estimate

$$\nu^\top \mathbb{E}\left[f(x+d)f(x+d)^\top\right]\nu = \mathbb{E}\left[\nu_1 \frac{x+d}{|x+d|} + \sum_{i=1}^{m-1} \nu_{i+1}\sin[i(x+d)]\right]^2.$$

First, we can calculate that

$$\mathbb{E}\left(\nu_1 \frac{x+d}{|x+d|}\right)^2 = \nu_1^2, \ \mathbb{E}\left[\nu_{i+1}\sin[i(x+d)]\right]^2 = \nu_{i+1}^2 \cdot \frac{1}{2}, \quad \forall i \in \{1, \ldots, m-1\}. \tag{41}$$

Then, for every $i \in \{1, \ldots, m-1\}$, we have

$$\mathbb{E}\left[\nu_1 \frac{x+d}{|x+d|} \cdot \nu_{i+1}\sin[i(x+d)]\right] \tag{42}$$

$$= \nu_1\nu_{i+1}\left[\int_{-|x|-2\pi}^{-|x|-\pi} \frac{x+d}{|x+d|}\sin[i(x+d)]\,\mathrm{d}d + \int_{|x|+\pi}^{|x|+2\pi} \frac{x+d}{|x+d|}\sin[i(x+d)]\,\mathrm{d}d\right]$$

$$= \nu_1\nu_{i+1}\left[\int_{-|x|-2\pi}^{-|x|-\pi} -\sin[i(x+d)]\,\mathrm{d}d + \int_{|x|+\pi}^{|x|+2\pi}\sin[i(x+d)]\,\mathrm{d}d\right] = 0.$$

For every $i, j \in \{1, \ldots, m-1\}$ such that $i \neq j$, it holds that

$$\mathbb{E}\left[\nu_{i+1}\sin[i(x+d)] \cdot \nu_{j+1}\sin[j(x+d)]\right] = \nu_{i+1}\nu_{j+1}\left[\int_{-|x|-2\pi}^{-|x|-\pi}\sin[i(x+d)]\sin[j(x+d)]\,\mathrm{d}d \right. \tag{43}$$

$$\left. + \int_{|x|+\pi}^{|x|+2\pi}\sin[i(x+d)]\sin[j(x+d)]\,\mathrm{d}d\right] = 0.$$

Combining equations (41)-(43), it follows that

$$\nu^\top \mathbb{E}\left[f(x+d)f(x+d)^\top\right]\nu = \nu_1^2 + \frac{1}{2}\sum_{i=1}^{m-1}\nu_{i+1}^2 \geq \frac{1}{2}\|\nu\|_2^2,$$

which implies that Assumption 1 holds with $\lambda^2 = 1/2$.

**Step 2.** In this step, we prove that the linear space spanned by the set of vectors

$$\mathcal{F}^c := \{f(x_t) \mid t \in \mathcal{K}^c\}$$

has dimension at most $m-1$ with probability at least $1 - \delta$. By the second property in (40), the subspace spanned by $\mathcal{F}^c$ is equivalent to that spanned by

$$\mathcal{F}' := \{f(x_t) \mid t \in \mathcal{K}'\},$$

where we define

$$\mathcal{K}' := \{t \mid t - 1 \in \mathcal{K}, \ t \in \mathcal{K}^c\}.$$

Therefore, the dimension of the subspace is at most $|\mathcal{K}'|$.

To estimate the cardinality of $\mathcal{K}'$, we divide $\mathcal{K}'$ into the following two disjoint sets:

$$\mathcal{K}'_1 := \{2t+1 \mid 2t \in \mathcal{K}, \ 2t+1 \in \mathcal{K}^c\}, \quad \mathcal{K}'_2 := \{2t \mid 2t-1 \in \mathcal{K}, \ 2t \in \mathcal{K}^c\}.$$

The size of $\mathcal{K}'_1$ is the summation of $\lceil T/2 \rceil$ independent Bernoulli random variables with parameter $p(1-p)$. Therefore, the Chernoff bound implies

$$\mathbb{P}\left[|\mathcal{K}'_1| \le 2p(1-p) \cdot \left\lceil \frac{T}{2} \right\rceil\right] \ge 1 - \exp\left[-\frac{p(1-p)}{3} \cdot \left\lceil \frac{T}{2} \right\rceil\right]. \tag{44}$$

Similarly, the size of $\mathcal{K}'_2$ is the summation of $\lfloor T/2 \rfloor$ independent Bernoulli random variables with parameter $p(1-p)$. Therefore, the Chernoff bound implies

$$\mathbb{P}\left[|\mathcal{K}'_2| \le 2p(1-p) \cdot \left\lfloor \frac{T}{2} \right\rfloor\right] \ge 1 - \exp\left[-\frac{p(1-p)}{3} \cdot \left\lfloor \frac{T}{2} \right\rfloor\right]. \tag{45}$$

Combining the bounds (44) and (45) and applying the union bound, it holds that

$$\mathbb{P}\left[|\mathcal{K}'| \le 2p(1-p)T\right] \ge 1 - \exp\left[-\frac{p(1-p)}{3} \cdot \left\lceil \frac{T}{2} \right\rceil\right] - \exp\left[-\frac{p(1-p)}{3} \cdot \left\lfloor \frac{T}{2} \right\rfloor\right]$$

$$\ge 1 - 2\exp\left[-\frac{p(1-p)T}{3}\right],$$

where the last inequality is because $\lfloor T/2 \rfloor \le \lceil T/2 \rceil \le T$. Since

$$T < \frac{m}{2p(1-p)},$$

we know

$$\mathbb{P}\left[|\mathcal{K}'| < m\right] \ge 1 - 2\exp\left(-m/3\right). \tag{46}$$

In addition, when $\mathcal{K}$ is the empty set $\emptyset$ or the full set $\{0, \ldots, T-1\}$, the set $\mathcal{K}'$ is an empty set, which implies that $|\mathcal{K}'|$ is smaller than $m$. This event happens with probability

$$p^\top + (1-p)^\top \ge 2[p(1-p)]^{T/2}.$$

Combining with inequality (46), we get

$$\mathbb{P}\left[|\mathcal{K}'| < m\right] \ge \max\left\{1 - 2\exp\left(-m/3\right), 2[p(1-p)]^{T/2}\right\}.$$

**Step 3.** Finally, we prove that if the dimension of the subspace spanned by $\mathcal{F}^c$ is smaller than $m$, the condition (8) cannot hold. Since the dimension of the subspace is at most $m-1$, there exists $Z \in \mathbb{R}^m$ such that

$$Zf(x_t) = 0, \quad \forall t \in \mathcal{K}^c.$$

With this choice of $Z$, the condition on the left-hand-side of (8) holds while the strict inequality on the right-hand-side fails. Therefore, we know that $\bar{A}$ is not the unique global solution to (4). $\square$

### B.7 Proof of Theorem 6

*Proof of Theorem 6.* The proof is similar to that of Theorem 4. Since both sides of inequality (9) are affine in $Z$, it suffices to prove that

$$\mathbb{P}\left[\hat{d}_1(Z) - \hat{d}_2(Z) < 0, \ \forall Z \in \mathbb{S}_F\right] \ge 1 - \delta,$$

where $\mathbb{S}_F$ is the Frobenius-norm unit sphere in $\mathbb{R}^{n \times m}$ and

$$\hat{d}_1(Z) := \sum_{t \in \mathcal{K}} \left\langle Z^\top, f(x_t)\hat{d}_t^\top \right\rangle, \quad \hat{d}_2(Z) := \sum_{t \in \mathcal{K}^c} \|Zf(x_t)\|_2.$$

The proof is divided into two steps.

**Step 1.**  First, we fix the vector $Z \in \mathbb{S}_F$ and prove that

$$\mathbb{P}\left[\hat{d}_1(Z) - \hat{d}_2(Z) < -\theta\right] \geq 1 - \delta,$$

holds for some constant $\theta > 0$. The proof of this step is divided into two steps.

**Step 1-1.**  We first analyze the term $\hat{d}_1(Z)$. For each $k \in \mathcal{K}$, we define the following disturbance vectors:

$$\bar{d}_t^k := \begin{cases} \bar{d}_t & \text{if } t \leq k, \\ \mathbf{0}_n & \text{otherwise,} \end{cases} \quad \forall t \in \{0, \dots, T-1\}.$$

Then, we define the trajectory generated by the above disturbance vectors:

$$x_0^k = \mathbf{0}_m, \quad x_{t+1}^k = \bar{A}f(x_t^k) + \bar{d}_t^k, \quad \forall t \in \{0, \dots, T-1\}.$$

Let

$$\mathcal{K} = \{k_1, \dots, k_{|\mathcal{K}|}\},$$

where the elements are sorted as $k_1 < k_2 < \cdots < k_{|\mathcal{K}|}$. Under the above definition, we know $x_t^{k_{|\mathcal{K}|}} = x_t$ for all $t$. We define

$$g_t^{k_j} := \begin{cases} f(x_t^{k_j}) - f(x_t^{k_{j-1}}) & \text{if } j > 1, \\ f(x_t^{k_1}) & \text{if } j = 1, \end{cases} \quad \forall j \in \{1, \dots, |\mathcal{K}|\}.$$

We note that $g_t^{k_j}$ is measurable on $\mathcal{F}_{k_j}$. Using these introduced notations, we can write $\hat{d}_1(Z)$ as

$$\hat{d}_1(Z) = \sum_{j=1}^{|\mathcal{K}|} \left\langle Z^\top, f(x_{k_j})\hat{d}_{k_j}^\top \right\rangle = \sum_{j=1}^{|\mathcal{K}|} \left\langle Z^\top, \sum_{\ell=1}^{j-1} g_{k_j}^{k_\ell}\hat{d}_{k_j}^\top \right\rangle = \sum_{\ell=1}^{|\mathcal{K}|} \sum_{j=\ell+1}^{|\mathcal{K}|} \hat{d}_{k_j}^\top Z g_{k_j}^{k_\ell}.$$

Then, Assumption 6 implies that $\bar{d}_t$ is sub-Gaussian with parameter $\sigma$ conditional on $\mathcal{F}_t$. Now, we estimate the expectation

$$\mathbb{E}\left[\exp\left[\nu\hat{d}_1(Z)\right]\right],$$

where $\nu \in \mathbb{R}$ is an arbitrary constant. First, for each $\ell \in \{1, \dots, |\mathcal{K}| - 1\}$, we estimate the following probability:

$$\mathbb{P}\left(\left|\sum_{j=\ell+1}^{|\mathcal{K}|} \hat{d}_{k_j}^\top Z g_{k_j}^{k_\ell}\right| \geq \epsilon \ \middle| \ \mathcal{F}_{k_\ell}\right).$$

Since $\hat{d}_{k_j}$ is a unit vector and $\|Z\|_F = 1$, we know

$$\left\|\hat{d}_{k_j}^\top Z\right\|_2 \leq \|\hat{d}_{k_j}^\top\|_2\|Z\|_2 \leq \|\hat{d}_{k_j}^\top\|_2\|Z\|_F = 1. \tag{47}$$

Moreover, we can estimate that

$$\begin{aligned}
\left\|g_{k_j}^{k_\ell}\right\|_2 &= \left\|f(x_{k_j}^{k_\ell}) - f(x_{k_j}^{k_{\ell-1}})\right\|_2 \leq L\left\|x_{k_j}^{k_\ell} - x_{k_j}^{k_{\ell-1}}\right\|_2 \\
&= L\left\|\bar{A}\left[f\left(x_{k_j-1}^{k_\ell}\right) - f\left(x_{k_j-1}^{k_{\ell-1}}\right)\right]\right\|_2 \leq \rho L\left\|f\left(x_{k_j-1}^{k_\ell}\right) - f\left(x_{k_j-1}^{k_{\ell-1}}\right)\right\|_2 \\
&\leq L(\rho L)\left\|x_{k_j-1}^{k_\ell} - x_{k_j-1}^{k_{\ell-1}}\right\|_2 \leq \cdots \leq L(\rho L)^{k_j-k_\ell-1}\left\|x_{k_\ell+1}^{k_\ell} - x_{k_\ell+1}^{k_{\ell-1}}\right\|_2 \\
&= L(\rho L)^{k_j-k_\ell-1}\|\bar{d}_{k_\ell}\|_2,
\end{aligned} \tag{48}$$

where the first inequality holds because $f$ has Lipschitz constant $L$, the second inequality is from $\|\bar{A}\|_2 \leq \rho$ and the last equality holds because

$$x_{k_\ell+1}^{k_\ell} = \bar{A}f\left(x_{k_\ell}^{k_\ell}\right) + \bar{d}_{k_\ell}, \quad x_{k_\ell+1}^{k_{\ell-1}} = \bar{A}f\left(x_{k_\ell}^{k_{\ell-1}}\right) = \bar{A}f\left(x_{k_\ell}^{k_\ell}\right).$$

By the sub-Gaussian assumption (Assumption 6), it holds that

$$\mathbb{P}\left(\|\bar{d}_{k_\ell}\|_2 \geq \eta \;\middle|\; \mathcal{F}_{k_\ell}\right) \leq 2\exp\left(-\frac{\eta^2}{2\sigma^2}\right), \quad \forall \eta \geq 0. \tag{49}$$

Combining inequalities (47)-(49), we get

$$\mathbb{P}\left(\left|\sum_{j=\ell+1}^{|\mathcal{K}|} \hat{d}_{k_j}^\top Z^\top g_{k_j}^{k_\ell}\right| \geq \epsilon \;\middle|\; \mathcal{F}_{k_\ell}\right) \leq \mathbb{P}\left(\sum_{j=\ell+1}^{|\mathcal{K}|} \left\|g_{k_j}^{k_\ell}\right\|_2 \geq \epsilon \;\middle|\; \mathcal{F}_{k_\ell}\right)$$

$$\leq \mathbb{P}\left(\sum_{j=\ell+1}^{|\mathcal{K}|} L(\rho L)^{k_j - k_\ell - 1}\|\bar{d}_{k_\ell}\|_2 \geq \epsilon \;\middle|\; \mathcal{F}_{k_\ell}\right)$$

$$\leq \mathbb{P}\left(\frac{L(\rho L)^{\Delta_j}}{1 - \rho L}\|\bar{d}_{k_\ell}\|_2 \geq \epsilon \;\middle|\; \mathcal{F}_{k_\ell}\right) \leq 2\exp\left[-\frac{(1-\rho L)^2\epsilon^2}{2\sigma^2 L^2(\rho L)^{2\Delta_j}}\right], \tag{50}$$

where $\Delta_j := k_j - k_{j-1} - 1$ and the second last inequality is from

$$\sum_{j=\ell+1}^{|\mathcal{K}|} L(\rho L)^{k_j - k_\ell - 1} < \sum_{i=\Delta_j}^{\infty} L(\rho L)^i = \frac{L(\rho L)^{\Delta_j}}{1 - \rho L}.$$

Since

$$\mathbb{E}\left(\sum_{j=\ell+1}^{|\mathcal{K}|} \hat{d}_{k_j}^\top Z g_{k_j}^{k_\ell} \;\middle|\; \mathcal{F}_{k_\ell}\right) = 0,$$

inequality (50) implies that the random variable $\sum_{j=\ell+1}^{|\mathcal{K}|} \hat{d}_{k_j}^\top Z^\top g_{k_j}^{k_\ell}$ is zero-mean and sub-Gaussian with parameter $\sigma L/(1 - \rho L)$ conditional on $\mathcal{F}_{k_\ell}$. By the property of sub-Gaussian random variables, we have

$$\mathbb{E}\left[\exp\left(\nu\sum_{j=\ell+1}^{|\mathcal{K}|} \hat{d}_{k_j}^\top Z g_{k_j}^{k_\ell}\right)\;\middle|\; \mathcal{F}_{k_\ell}\right] \leq \exp\left[\frac{\nu^2\sigma^2 L^2(\rho L)^{2\Delta_j}}{2(1-\rho L)^2}\right], \quad \forall \nu \geq 0.$$

Finally, utilizing the tower property of conditional expectation, we have

$$\mathbb{E}\left[\exp\left[\nu\hat{d}_1(Z)\right]\right] = \mathbb{E}\left[\exp\left(\nu\sum_{\ell=1}^{|\mathcal{K}|-2}\sum_{j=\ell+1}^{|\mathcal{K}|} \hat{d}_{k_j}^\top Z g_{k_j}^{k_\ell}\right) \times \mathbb{E}\left[\exp\left(\nu\sum_{j=|\mathcal{K}|}^{|\mathcal{K}|} \hat{d}_{k_j}^\top Z g_{k_j}^{k_\ell}\right)\;\middle|\; \mathcal{F}_{k_{|\mathcal{K}|-1}}\right]\right] \tag{51}$$

$$\leq \mathbb{E}\left[\exp\left(\nu\sum_{\ell=1}^{|\mathcal{K}|-2}\sum_{j=\ell+1}^{|\mathcal{K}|} \hat{d}_{k_j}^\top Z g_{k_j}^{k_\ell}\right) \times \exp\left[\frac{\nu^2\sigma^2 L^2(\rho L)^{2\Delta_j}}{2(1-\rho L)^2}\right]\right]$$

$$\leq \cdots \leq \exp\left[\frac{\nu^2\sigma^2 L^2}{2(1-\rho L)^2}\sum_{j\in\mathcal{K}}(\rho L)^{2\Delta_j}\right], \quad \forall \nu \geq 0.$$

Since the random variable $(\rho L)^{\Delta_j}$ is bounded in $[0,1]$ and thus, it is sub-Gaussian with parameter $1/2$. Therefore, with a constant number of samples, the mean of $(\rho L)^{2\Delta_j}$ will concentrate around its expectation, which is approximately

$$\sum_{\Delta=0}^{\infty} p(1-p)^{2\Delta}(\rho L)^{2\Delta} = \frac{p}{1 - (1-p)^2(\rho L)^2} \leq \frac{p}{1 - \rho L}.$$

Then, the bound in (51) becomes

$$\mathbb{E}\left[\exp\left[\nu\hat{d}_1(Z)\right]\right] \lesssim \exp\left[\frac{\nu^2\sigma^2L^2p|\mathcal{K}|}{2(1-\rho L)^3}\right], \quad \forall \nu \geq 0. \tag{52}$$

Applying Chernoff's bound to (52), we get

$$\mathbb{P}\left[\hat{d}_1(Z) \leq \epsilon\right] \geq 1 - \exp\left[-\frac{(1-\rho L)^3}{2\sigma^2L^2p|\mathcal{K}|}\cdot\epsilon^2\right], \quad \forall \epsilon \geq 0. \tag{53}$$

**Step 1-2.** Next, we analyze the term $\hat{d}_2(Z)$. Define the set

$$\mathcal{K}' := \{t \mid 1 \leq t \leq T, \ t \in \mathcal{K}^c, \ t-1 \in \mathcal{K}\}.$$

With probability at least $1 - \exp[-\Theta[p(1-p)T]]$, we have

$$|\mathcal{K}'| = \Theta[p(1-p)T].$$

Therefore, $\mathcal{K}'$ is non-empty with high probability. Since $\|Zf(x_t)\|_2 \geq 0$ for all $t \in \mathcal{K}^c$, we know

$$\hat{d}_2(Z) \geq \sum_{k \in \mathcal{K}'} \|Zf(x_t)\|_2.$$

To establish a high-probability lower bound of $\|Zf(x_t)\|_2$, we prove the following lemma.

**Lemma 3.** *For each $t \in \mathcal{K}'$, it holds that*

$$\mathbb{P}\left[\|Zf(x_t)\|_2 \geq \frac{\lambda}{2} \ \middle| \ \mathcal{F}_t\right] \geq \frac{c\lambda^4}{\sigma^4L^4},$$

*where $c := 1/1058$ is an absolute constant.*

For each $t \in \mathcal{K}'$, let $\mathbf{1}_t$ be the indicator of the event that $\|Zf(x_t)\|_2$ is larger than the $\frac{c\lambda^4}{\sigma^4L^4}$-quantile conditional on $\mathcal{F}_t$. Then, it holds that

$$\mathbb{P}(\mathbf{1}_t = 1 \mid \mathcal{F}_t) = 1 - \mathbb{P}(\mathbf{1}_t = 0 \mid \mathcal{F}_t) = \frac{c\lambda^4}{\sigma^4L^4}.$$

Therefore, we know

$$\left\{\mathbf{1}_t - \frac{c\lambda^4}{\sigma^4L^4}, \ t \in \mathcal{K}'\right\}$$

is a martingale with respect to filtration set $\{\mathcal{F}_t, \ t \in \mathcal{K}'\}$. Applying Azuma's inequality, it holds with probability at least $1 - \exp[-\Theta(\frac{\lambda^4|\mathcal{K}'|}{\sigma^4L^4})]$ that

$$\sum_{t \in \mathcal{K}'} \mathbf{1}_t \geq \frac{c\lambda^4|\mathcal{K}'|}{2\sigma^4L^4},$$

which means that for at least $\frac{c\lambda^4|\mathcal{K}'|}{2\sigma^4L^4}$ elements in $\mathcal{K}'$, the event that $\|Zf(x_t)\|_2$ is larger than the $\frac{c\lambda^4}{\sigma^4L^4}$-quantile conditional on $\mathcal{F}_t$ happens. Using the lower bound on the quantile in Lemma 3, we know

$$\sum_{t \in \mathcal{K}'} \|Zf(x_t)\|_2 \geq \frac{c\lambda^4|\mathcal{K}'|}{2\sigma^4L^4}\cdot\frac{\lambda}{2} + \left(|\mathcal{K}'| - \frac{c\lambda^4|\mathcal{K}'|}{2\sigma^4L^4}\right)\cdot 0 = \frac{c\lambda^5|\mathcal{K}'|}{4\sigma^4L^4} \tag{54}$$

holds with the same probability.

Combining inequalities (53) and (54), we get

$$\mathbb{P}\left[f(Z) \leq \epsilon - \frac{c\lambda^5|\mathcal{K}'|}{4\sigma^4L^4}\right] \geq 1 - \exp\left[-\frac{(1-\rho L)^3}{2\sigma^2L^2p|\mathcal{K}|}\cdot\epsilon^2\right] - \exp\left[-\Theta\left(\frac{\lambda^4|\mathcal{K}'|}{\sigma^4L^4}\right)\right],$$

where we define $f(Z) := \hat{d}_1(Z) - \hat{d}_2(Z)$. Choosing

$$\epsilon := \frac{c\lambda^5|\mathcal{K}'|}{8\sigma^4 L^4},$$

it follows that

$$\mathbb{P}\left[f(Z) \le -\frac{c\lambda^5|\mathcal{K}'|}{8\sigma^4 L^4}\right] \ge 1 - \exp\left[-\Theta\left(\frac{(1-\rho L)^3\lambda^{10}|\mathcal{K}'|^2}{\sigma^{10}L^{10}p|\mathcal{K}|}\right)\right] - \exp\left[-\Theta\left(\frac{\lambda^4|\mathcal{K}'|}{\sigma^4 L^4}\right)\right]. \tag{55}$$

By the definition of the probabilistic sparsity model, it holds with probability at least $1 - \exp[-\Theta[p(1-p)T]]$ that

$$|\mathcal{K}| \le 2pT, \quad |\mathcal{K}'| \ge \frac{p(1-p)T}{2}. \tag{56}$$

Therefore, the probability bound in (55) becomes

$$\mathbb{P}\left[f(Z) \le -\frac{c\lambda^5 p(1-p)T}{16\sigma^4 L^4}\right] \ge 1 - \exp\left[-\Theta\left(\frac{(1-\rho L)^3\lambda^{10}(1-p)^2 T}{\sigma^{10}L^{10}}\right)\right]$$
$$- \exp\left[-\Theta\left(\frac{\lambda^4 p(1-p)T}{\sigma^4 L^4}\right)\right] - \exp[-\Theta[p(1-p)T]].$$

Now, if the sample complexity satisfies

$$T = \Omega\left[\max\left\{\frac{\kappa^{10}}{(1-\rho L)^3(1-p)^2}, \frac{\kappa^4}{p(1-p)}\right\}\log\left(\frac{1}{\delta}\right)\right], \tag{57}$$

we know

$$\mathbb{P}\left[f(Z) \le -\theta\right] \ge 1 - \delta, \tag{58}$$

where we define

$$\kappa := \frac{\sigma L}{\lambda}, \quad \theta := \frac{c\lambda^5 p(1-p)T}{16\sigma^4 L^4}.$$

**Step 2.** In the second step, we apply discretization techniques to prove that condition (58) holds for all $Z \in \mathbb{S}_F$. For a sufficiently small constant $\epsilon > 0$, let

$$\{Z^1, \ldots, Z^N\}$$

be an $\epsilon$-cover of the unit ball $\mathbb{S}_F$. Namely, for all $Z \in \mathbb{S}_F$, we can find $r \in \{1, 2, \ldots, N\}$ such that $\|Z - Z^r\|_F \le \epsilon$. It is proved in Wainwright (2019) that the number of points $N$ can be bounded by

$$\log(N) \le mn\log\left(1 + \frac{2}{\epsilon}\right).$$

Now, we estimate the Lipschitz constant of $f(Z)$ and construct a high-probability upper bound for the Lipschitz constant. For all $Z, Z' \in \mathbb{R}^{n \times m}$, we can calculate that

$$f(Z) - f(Z') = \sum_{t\in\mathcal{K}}\left\langle(Z - Z')^\top, f(x_t)\hat{d}_t^\top\right\rangle - \sum_{t\in\mathcal{K}^c}\left(\|Zf(x_t)\|_2 - \|Z'f(x_t)\|_2\right)$$
$$\le \|Z - Z'\|_F\sum_{t\in\mathcal{K}}\left\|f(x_t)\hat{d}_t^\top\right\|_F + \|Z - Z'\|_2\sum_{t\in\mathcal{K}^c}\|f(x_t)\|_2$$
$$\le \|Z - Z'\|_F\sum_{t=0}^{T-1}\|f(x_t)\|_2. \tag{59}$$

Using the decomposition in **Step 1-1**, we have

$$f(x_t) = \sum_{\ell=1}^{j} g_t^{k_\ell},$$

where $k_j$ is the maximal element in $\mathcal{K}$ such that $k_j < t$. Therefore, we can calculate that

$$\sum_{t=0}^{T-1} \|f(x_t)\|_2 \leq \sum_{j=1}^{|\mathcal{K}|} \sum_{t=k_j+1}^{T-1} \left\| g_t^{k_j} \right\|_2. \tag{60}$$

For each $j \in \{1, \ldots, |\mathcal{K}|\}$, we can prove in the same way as (48) that

$$\left\| g_t^{k_j} \right\|_2 \leq L(\rho L)^{k_j - t - 1} \|\bar{d}_{k_j}\|_2, \quad \forall t > k_j.$$

Substituting into inequality (60), it follows that

$$\sum_{t=0}^{T-1} \|f(x_t)\|_2 \leq \sum_{j=1}^{|\mathcal{K}|} \sum_{t=k_j+1}^{T-1} L(\rho L)^{k_j - t - 1} \|\bar{d}_{k_j}\|_2 \leq \frac{L}{1 - \rho L} \sum_{j=1}^{|\mathcal{K}|} \|\bar{d}_{k_j}\|_2.$$

Using Assumption 6 and the same technique as in (51), we know

$$\mathbb{P}\left( \sum_{j=1}^{|\mathcal{K}|} \|\bar{d}_{k_j}\|_2 \leq \eta \right) \geq 1 - 2\exp\left( -\frac{\eta^2}{2\sigma^2 |\mathcal{K}|} \right) \geq 1 - 2\exp\left( -\frac{\eta^2}{4\sigma^2 pT} \right),$$

where the second inequality is from the high probability bound in (56). Hence, it holds that

$$\mathbb{P}\left( \sum_{t=0}^{T-1} \|f(x_t)\|_2 \leq \eta \right) \geq 1 - 2\exp\left( -\frac{\eta^2 (1 - \rho L)^2}{4\sigma^2 L^2 pT} \right), \tag{61}$$

Choosing

$$\eta := \frac{\theta}{2\epsilon},$$

the bound in (61) becomes

$$\mathbb{P}\left( \sum_{t=0}^{T-1} \|f(x_t)\|_2 \leq \frac{\theta}{2\epsilon} \right) \geq 1 - 2\exp\left( -\frac{(1 - \rho L)^2}{4\sigma^2 L^2 pT \epsilon^2} \cdot \theta^2 \right) \tag{62}$$

$$= 1 - 2\exp\left[ -\Theta\left[ \frac{(1 - \rho L)^2}{4\sigma^2 L^2 pT \epsilon^2} \cdot \left( \frac{\lambda^5 p(1-p)T}{\sigma^4 L^4} \right)^2 \right] \right]$$

$$= 1 - 2\exp\left[ -\Theta\left[ \frac{(1 - \rho L)^2 \kappa^{10} p(1-p)^2 T}{\epsilon^2} \right] \right].$$

We set

$$\epsilon := \Theta\left[ \sqrt{(1 - \rho L)^2 \kappa^{10} p(1-p)^2} \right].$$

Then, it follows that

$$\exp\left[ -\Theta\left[ \frac{(1 - \rho L)^2 \kappa^{10} p(1-p)^2 T}{\epsilon^2} \right] \right] = \exp\left[ -\Theta(T) \right] \leq \frac{\delta}{4},$$

where the last inequality is from the choice of $T$ in (57). Substituting back into (62), we get

$$\mathbb{P}\left(\sum_{t=0}^{T-1}\|f(x_t)\|_2 \leq \frac{\theta}{2\epsilon}\right) \geq 1 - \frac{\delta}{2}. \tag{63}$$

Under the event in (63), for all $Z \in \mathbb{S}_F$, there exists an element $Z^r$ in the $\epsilon$-net such that

$$f(Z) \leq f(Z^r) + \epsilon \cdot \sum_{t=0}^{T-1}\|f(x_t)\|_2 \leq f(Z^r) + \frac{\theta}{2}.$$

If we replace $\delta$ with $\delta/(2N)$ in (58) and choose $Z = Z^r$ for all $r \in \{1, \ldots, N\}$, the union bound implies that

$$\mathbb{P}\left[f(Z^r) \leq -\theta, \ r = 1, \ldots, N\right] \geq 1 - \frac{\delta}{2}. \tag{64}$$

Under the above conditions, we have

$$f(Z) \leq f(Z^r) + \frac{\theta}{2} \leq -\frac{\theta}{2} < 0.$$

To satisfy condition (64), the sample complexity bound (57) becomes

$$T = \Omega\left[\max\left\{\frac{\kappa^{10}}{(1-\rho L)^3(1-p)^2}, \frac{\kappa^4}{p(1-p)}\right\}\log\left(\frac{2N}{\delta}\right)\right]$$

$$= \Theta\left[\max\left\{\frac{\kappa^{10}}{(1-\rho L)^3(1-p)^2}, \frac{\kappa^4}{p(1-p)}\right\} \times \left[mn\log\left(\frac{1}{(1-\rho L)\kappa p(1-p)}\right) + \log\left(\frac{1}{\delta}\right)\right]\right],$$

which is the desired sample complexity bound in the theorem.

**Lower bound of $\kappa$.** Before we close the proof, we provide a lower bound of $\kappa = \sigma L/\lambda$. Equivalently, we provide an upper bound on $\lambda^2$, which is at most the minimal eigenvalue of

$$\mathbb{E}\left[f(x + \bar{d}_t)f(x + \bar{d}_t)^\top \mid \mathcal{F}_t, \bar{d}_t \neq \mathbf{0}_n\right].$$

Let $\nu \in \mathbb{R}^m$ be a vector satisfying

$$\|\nu\|_2 = 1, \quad \nu^\top f(x) = 0.$$

Then, we know

$$\nu^\top f(x + \bar{d}_t)f(x + \bar{d}_t)^\top \nu = \nu^\top\left[f(x + \bar{d}_t) - f(x)\right]\left[f(x + \bar{d}_t) - f(x)\right]^\top \nu \tag{65}$$

$$= \left[\left[f(x + \bar{d}_t) - f(x)\right]^\top \nu\right]^2 \leq \left\|f(x + \bar{d}_t) - f(x)\right\|_2^2$$

$$\leq L^2\|\bar{d}_t\|_2^2,$$

where the last inequality is from the Lipschitz continuity of $f$. Using the sub-Gaussian assumption, it follows that

$$\mathbb{E}\left[\|\bar{d}_t\|_2^2 \mid \mathcal{F}_t, \ \bar{d}_t \neq \mathbf{0}_n\right] \leq \sigma^2, \tag{66}$$

where we utilize the fact that the standard deviation of sub-Gaussian random variables with parameter $\sigma$ is at most $\sigma$. Combining inequalities (65) and (66), it follows that

$$\nu^\top \mathbb{E}\left[f(x + \bar{d}_t)f(x + \bar{d}_t)^\top \mid \mathcal{F}_t, \bar{d}_t \neq \mathbf{0}_n\right]\nu \leq \sigma^2 L^2.$$

Therefore, it holds that

$$\lambda^2 \leq \lambda_{min}\left[\mathbb{E}\left[f(x + \bar{d}_t)f(x + \bar{d}_t)^\top \mid \mathcal{F}_t, \bar{d}_t \neq \mathbf{0}_n\right]\right] \leq \sigma^2 L^2, \quad \forall x \in \mathbb{R}^n,$$

which further leads to

$$\kappa = \frac{\sigma L}{\lambda} \geq 1.$$

This completes the proof. $\qquad\square$

### B.8 Proof of Lemma 3

*Proof of Lemma 3.* Let

$$\delta := \frac{c\lambda^4}{\sigma^4 L^4}, \quad \theta_t := \left\| Z^\top f \left[ \bar{A} f(x_{t-1}) \right] \right\|_2.$$

We finish the proof by discussing two cases.

**Case 1.** We first consider the case when

$$\theta_t \geq \frac{\lambda}{2} + \sqrt{2\sigma^2 L^2 \log\left( \frac{2}{1-\delta} \right)}.$$

Using the Lipschitz continuity of $f$, we have

$$
\begin{aligned}
\| Zf(x_t) \|_2 &= \left\| \left[ Zf(x_t) - Z^\top f \left[ \bar{A} f(x_{t-1}) \right] \right] + Zf \left[ \bar{A} f(x_{t-1}) \right] \right\|_2 \\
&\geq \left\| Zf \left[ \bar{A} f(x_{t-1}) \right] \right\|_2 - \left\| Zf(x_t) - Zf \left[ \bar{A} f(x_{t-1}) \right] \right\|_2 \\
&\geq \theta_t - \| Z \|_2 \left\| f(x_t) - f \left[ \bar{A} f(x_{t-1}) \right] \right\|_2 \\
&\geq \theta_t - \| Z \|_F \cdot L \left\| \bar{d}_t \right\|_2 \geq \theta_t - L \left\| \bar{d}_t \right\|_2.
\end{aligned}
\tag{67}
$$

By Assumption 6, we know $\left\| \bar{d}_t \right\|_2 = |\ell_t|$ and it follows that

$$\mathbb{P}\left( \left\| \bar{d}_t \right\|_2 \geq \epsilon \mid \mathcal{F}_t \right) \leq 2 \exp\left( -\frac{\epsilon^2}{2\sigma^2} \right), \quad \forall \epsilon \geq 0.$$

Therefore, we get the estimation

$$
\begin{aligned}
\mathbb{P}\left( \| Zf(x_t) \|_2 \leq \frac{\lambda}{2} \;\middle|\; \mathcal{F}_t \right) &\leq \mathbb{P}\left( \theta_t - L \left\| \bar{d}_t \right\|_2 \leq \frac{\lambda}{2} \;\middle|\; \mathcal{F}_t \right) \\
&= \mathbb{P}\left( \left\| \bar{d}_t \right\|_2 \geq \frac{\theta_t - \lambda/2}{L} \;\middle|\; \mathcal{F}_t \right) \\
&\leq \mathbb{P}\left( \left\| \bar{d}_t \right\|_2 \geq \sqrt{2\sigma^2 \log\left( \frac{2}{1-\delta} \right)} \;\middle|\; \mathcal{F}_t \right) \leq 1 - \delta.
\end{aligned}
$$

Therefore, we have proved that

$$\mathbb{P}\left( \| Zf(x_t) \|_2 \geq \frac{\lambda}{2} \;\middle|\; \mathcal{F}_t \right) \geq \delta.$$

**Case 2.** Then, we focus on the case when

$$\theta_t \leq \frac{\lambda}{2} + \sqrt{2\sigma^2 L^2 \log\left( \frac{2}{1-\delta} \right)}.$$
$$\tag{68}$$

Assume conversely that

$$\mathbb{P}\left( \| Zf(x_t) \|_2 \geq \frac{\lambda}{2} \;\middle|\; \mathcal{F}_t \right) < \delta.$$
$$\tag{69}$$

Similar to inequality (67), the Lipschitz continuity of $f$ implies

$$\| Zf(x_t) \|_2 \leq \theta_t + L \left\| \bar{d}_t \right\|_2.$$

Therefore, by applying Assumption 6, we get the tail bound

$$\mathbb{P}\left(\|Zf(x_t)\|_2 \geq \theta \mid \mathcal{F}_t\right) \leq \mathbb{P}\left(\theta_t + L\left\|\bar{d}_t\right\|_2 \geq \theta \mid \mathcal{F}_t\right)$$
$$=\mathbb{P}\left(\left\|\bar{d}_t\right\|_2 \geq \frac{\theta - \theta_t}{L} \ \Big| \ \mathcal{F}_t\right) \leq 2\exp\left[-\frac{(\theta - \theta_t)^2}{2\sigma^2 L^2}\right], \quad \forall \theta \geq \theta_t.$$

Define $(x)_+ := \max\{x, 0\}$. The above bound leads to

$$\mathbb{P}\left(\|Zf(x_t)\|_2 \geq \theta \mid \mathcal{F}_t\right) \leq 2\exp\left[-\frac{(\theta - \theta_t)_+^2}{2\sigma^2 L^2}\right], \quad \forall \theta \in \mathbb{R}. \tag{70}$$

Using the definition of expectation, we can calculate that

$$\mathbb{E}\left[\|Zf(x_t)\|_2^2 \mid \mathcal{F}_t\right] = \int_0^\infty 2\theta \cdot \mathbb{P}\left[\|Zf(x_t)\|_2 \geq \theta \mid \mathcal{F}_t\right] \ d\theta$$
$$\leq \frac{\lambda^2}{4} + \int_{\lambda/2}^\infty 2\theta \cdot \mathbb{P}\left[\|Zf(x_t)\|_2 \geq \theta \mid \mathcal{F}_t\right] \ d\theta.$$

By condition (69), we get

$$\mathbb{P}\left[\|Zf(x_t)\|_2 \geq \theta \mid \mathcal{F}_t\right] \leq \mathbb{P}\left[\|Zf(x_t)\|_2 \geq \frac{\lambda}{2} \ \Big| \ \mathcal{F}_t\right] \leq \delta, \quad \forall \theta \geq \frac{\lambda}{2}.$$

Combining with inequality (70), it follows that

$$\mathbb{E}\left[\|Zf(x_t)\|_2^2 \mid \mathcal{F}_t\right] \leq \frac{\lambda^2}{4} + \int_{\lambda/2}^\infty 2\theta \cdot \min\left\{\delta, 2\exp\left[-\frac{(\theta - \theta_t)_+^2}{2\sigma^2 L^2}\right]\right\} \ d\theta \tag{71}$$
$$= \frac{\lambda^2}{4} + \delta\left(\theta_1^2 - \frac{\lambda^2}{4}\right) + \int_{\theta_1}^\infty 4\theta \exp\left[-\frac{(\theta - \theta_t)^2}{2\sigma^2 L^2}\right] \ d\theta,$$

where we define

$$\theta_1 := \max\left\{\frac{\lambda}{2}, \theta_t + \sqrt{2\sigma^2 L^2 \log\left(\frac{2}{\delta}\right)}\right\} \geq \theta_t.$$

Using condition (68), we know

$$\theta_1^2 \leq \left(\frac{\lambda}{2} + \sqrt{2\sigma^2 L^2 \log\left(\frac{2}{1-\delta}\right)} + \sqrt{2\sigma^2 L^2 \log\left(\frac{2}{\delta}\right)}\right)^2 \tag{72}$$
$$\leq \left(\frac{\lambda}{2} + 2\sqrt{2\sigma^2 L^2 \log\left(\frac{2}{\delta}\right)}\right)^2 \leq \frac{\lambda^2}{2} + 16\sigma^2 L^2 \log\left(\frac{2}{\delta}\right),$$

where the last inequality is from Cauchy's inequality. Moreover, we can estimate that

$$\int_{\theta_1}^\infty 4\theta \exp\left[-\frac{(\theta - \theta_t)^2}{2\sigma^2 L^2}\right] \ d\theta \leq \int_{\theta_2}^\infty 4\theta \exp\left[-\frac{(\theta - \theta_t)^2}{2\sigma^2 L^2}\right] \ d\theta \tag{73}$$
$$= \int_{\theta_2}^\infty 4\theta_t \exp\left[-\frac{(\theta - \theta_t)^2}{2\sigma^2 L^2}\right] \ d\theta + \int_{\theta_2}^\infty 4(\theta - \theta_t) \exp\left[-\frac{(\theta - \theta_t)^2}{2\sigma^2 L^2}\right] \ d\theta$$
$$= \int_{\theta_2}^\infty 4\theta_t \exp\left[-\frac{(\theta - \theta_t)^2}{2\sigma^2 L^2}\right] \ d\theta + 2\delta\sigma^2 L^2,$$

where we denote $\theta_2 := \theta_t + \sqrt{2\sigma^2 L^2 \log\left(\frac{2}{\delta}\right)} \leq \theta_1$. Utilizing the following bound on the cumulative density function of the standard Gaussian distribution:

$$\int_\eta^\infty e^{-\frac{x^2}{2}} \ dx \leq \eta^{-1} e^{-\frac{\eta^2}{2}}, \quad \forall \eta > 0,$$

we have

$$\int_{\theta_2}^{\infty} 4\theta_t \exp\left[-\frac{(\theta-\theta_t)^2}{2\sigma^2 L^2}\right] \, d\theta \leq 4\theta_t \sigma L \cdot \frac{1}{\sqrt{2\log\left(\frac{2}{\delta}\right)}} \cdot \frac{\delta}{2} \leq \sqrt{2}\theta_t \cdot \delta\sigma L.$$

Combining with (73), it follows that

$$\int_{\theta_1}^{\infty} 4\theta \exp\left[-\frac{(\theta-\theta_t)^2}{2\sigma^2 L^2}\right] \, d\theta \leq \sqrt{2}\theta_t \cdot \delta\sigma L + 2\delta\sigma^2 L^2 \leq 4\delta\theta_t^2 + 4\delta\sigma^2 L^2, \tag{74}$$

where the last inequality is from Cauchy's inequality. Substituting inequalities (72) and (74) back into (71), we get

$$\mathbb{E}\left[\|Zf(x_t)\|_2^2 \mid \mathcal{F}_t\right] \leq \frac{\lambda^2}{4} + \delta\left[\frac{\lambda^2}{4} + 16\sigma^2 L^2 \log\left(\frac{2}{\delta}\right)\right] + 4\delta\theta_t^2 + 4\delta\sigma^2 L^2$$

$$\leq \frac{(1+\delta)\lambda^2}{4} + 16\sigma^2 L^2 \cdot \delta\log\left(\frac{2}{\delta}\right) + \delta\left[\frac{\lambda}{2} + \sqrt{2\sigma^2 L^2 \log\left(\frac{2}{1-\delta}\right)}\right]^2 + 4\delta\sigma^2 L^2$$

$$\leq \frac{(1+\delta)\lambda^2}{4} + 16\sigma^2 L^2 \cdot \delta\log\left(\frac{2}{\delta}\right) + \frac{\delta\lambda^2}{2} + 4\sigma^2 L^2 \cdot \delta\log\left(\frac{2}{\delta}\right) + 4\delta\sigma^2 L^2$$

$$\leq \frac{(1+3\delta)\lambda^2}{4} + 24\sigma^2 L^2 \cdot \delta\log\left(\frac{2}{\delta}\right).$$

where the second inequality is from (68) and the last inequality is from Cauchy's inequality and $\delta < 1/2$. On the other hand, Assumption 1 implies that

$$\mathbb{E}\left(\|Zf(x_t)\|_2^2 \mid \mathcal{F}_t\right) = \left\langle ZZ^\top, \mathbb{E}\left[f(x_t)f(x_t)^\top \mid \mathcal{F}_t\right]\right\rangle \geq \lambda^2 \|Z\|_F^2 = \lambda^2.$$

Combining the last two inequalities, we get

$$\lambda^2 \leq \frac{(1+3\delta)\lambda^2}{4} + 24\sigma^2 L^2 \cdot \delta\log\left(\frac{2}{\delta}\right),$$

which is equivalent to

$$\delta\log\left(\frac{2}{\delta}\right) \geq \frac{(3-3\delta)\lambda^2}{96\sigma^2 L^2} \geq \frac{\lambda^2}{23\sigma^2 L^2}.$$

For all $x \in (0,1)$, it holds that $x\log(2/x) < \sqrt{2x}$. Hence, we have

$$\sqrt{2\delta} > \frac{\lambda^2}{23\sigma^2 L^2},$$

which contradicts with our assumption (69).

$\square$

## B.9 Proof of Theorem 7

*Proof of Theorem 7.* In this proof, we focus on the case when $m = n$, and the counterexample can be easily extended into more general cases. We construct the following system dynamics:

$$\bar{A} := \rho \boldsymbol{I}_n, \quad f(x) := x, \quad \forall x \in \mathbb{R}^n,$$

where $\rho \geq 2+\sqrt{6}$ is a constant. One can verify Assumption 4 holds with Lipschitz constant $L = 1$. Therefore, the stability condition (Assumption 5) is violated since $\rho > 1/L$. The system dynamics can be written as

$$x_t = \sum_{k \in \mathcal{K}, k < t} \rho^{t-k-1} d_k, \quad \forall t \in \{0, \ldots, T\}. \tag{75}$$

Conditional on $\mathcal{F}_t$ and $t \in \mathcal{K}$, the disturbance vector is generated as

$$d_t \sim \text{Uniform}(\mathbb{S}^{n-1}),$$

where $\mathbb{S}^{n-1}$ is the unit ball $\{d \in \mathbb{R}^n \mid \|d\|_2 = 1\}$. The attack/disturbance model satisfies Assumption 1 with $\lambda = 1/\sqrt{n}$ and Assumption 6 with $\sigma = 1/\sqrt{n}$. Define the event

$$\mathcal{E} := \{T - 1 \in \mathcal{K}, |\mathcal{K}| > 1\}.$$

By the definition of the probabilistic sparsity model, we can calculate that

$$\mathbb{P}(\mathcal{E}) = p\left[1 - (1-p)^{T-1}\right].$$

Our goal is to prove that

$$\mathbb{P}\left[\hat{d}_1(Z) - \hat{d}_2(Z) > 0 \mid \mathcal{E}\right] = 1,$$

where we define

$$\hat{d}_1(Z) := \sum_{t \in \mathcal{K}} \left\langle Z^\top, f(x_t)\hat{d}_t^\top \right\rangle, \quad \hat{d}_2(Z) := \sum_{t \in \mathcal{K}^c} \|Zf(x_t)\|_2.$$

Then, by Theorem 1, we know that $\bar{A}$ is not a global solution to problem (4) with probability at least

$$p\left[1 - (1-p)^{T-1}\right].$$

Let $t_1$ be the smallest element in $\mathcal{K}$, namely, the first time instance when there is a nonzero disturbance. Under event $\mathcal{E}$, it holds that $t_1 < T - 1$. We first prove that

$$x_t \neq \mathbf{0}_n, \quad \forall t \in \{t_1 + 1, \ldots, T - 1\}.$$

By the system dynamics (75) and the triangle inequality, we have

$$\|x_t\|_2 \geq \rho^{t-t_1-1}\|d_{t_1}\|_2 - \sum_{k \in \mathcal{K}, t_1 < k < t} \rho^{t-k-1}\|d_k\|_2 = \rho^{t-t_1-1} - \sum_{k \in \mathcal{K}, t_1 < k < t} \rho^{t-k-1}$$

$$\geq \rho^{t-t_1-1} - \sum_{i=0}^{t-t_1-2} \rho^i = \frac{\rho^{t-t_1} - 2\rho^{t-t_1-1} + 1}{\rho - 1} > 0,$$

where the last inequality holds because $\rho \geq 2$. Then, we choose

$$Z := x_{T-1}\hat{d}_{T-1}^\top \neq 0.$$

It follows that

$$\hat{d}_1(Z) = \sum_{t \in \mathcal{K}} \left\langle Z^\top, f(x_t)\hat{d}_t^\top \right\rangle = \left\| x_{T-1}\hat{d}_{T-1}^\top \right\|_F^2 + \sum_{t \in \mathcal{K}, t < T-1} \left\langle x_{T-1}\hat{d}_{T-1}^\top, f(x_t)\hat{d}_t^\top \right\rangle$$

$$\geq \|x_{T-1}\|_2^2 - \sum_{t \in \mathcal{K}, t < T-1} \|x_{T-1}\|_2 \|x_t\|_2,$$

$$\hat{d}_2(Z) = \sum_{t \in \mathcal{K}^c} \|Zf(x_t)\|_2 = \sum_{t \in \mathcal{K}^c} \left\| x_{T-1}\hat{d}_{T-1}^\top x_t \right\|_2 \leq \sum_{t \in \mathcal{K}^c} \|x_{T-1}\|_2 \|x_t\|_2.$$

Combining the above two inequalities, we get

$$\hat{d}_1(Z) - \hat{d}_2(Z) \leq \|x_{T-1}\|_2 \left( \|x_{T-1}\|_2 - \sum_{t=0}^{T-2} \|x_t\|_2 \right) = \|x_{T-1}\|_2 \left( \|x_{T-1}\|_2 - \sum_{t=t_1+1}^{T-2} \|x_t\|_2 \right),$$

where the last equality holds because $x_t = \mathbf{0}_n$ for all $t \leq t_1$. Since $\|x_{T-1}\|_2 > 0$, it is sufficient to prove that

$$\|x_{T-1}\|_2 > \sum_{t=t_1+1}^{T-2} \|x_t\|_2. \tag{76}$$

Considering the system dynamics (75) and the fact that $\|d_k\|_2 = 1$ for all $k \in \mathcal{K}$ , we have the estimation

$$\rho^{t-t_1-1} - \sum_{k \in \mathcal{K}, t_1 < k < t} \rho^{t-k-1} \leq \|x_t\|_2 \leq \sum_{k \in \mathcal{K}, k < t} \rho^{t-k-1}.$$

The desired inequality (76) holds if we can show

$$\rho^{T-1-t_1-1} - \sum_{k \in \mathcal{K}, t_1 < k < T-1} \rho^{T-1-k-1} > \sum_{t=t_1+1}^{T-2} \sum_{k \in \mathcal{K}, k < t} \rho^{t-k-1},$$

which is further equivalent to

$$2\rho^{T-t_1-2} > \sum_{t=t_1+1}^{T-1} \sum_{k \in \mathcal{K}, k < t} \rho^{t-k-1}$$

$$\Longleftarrow 2\rho^{T-t_1-2} > \sum_{t=t_1+1}^{T-1} \sum_{k=t_1}^{t-1} \rho^{t-k-1} = \sum_{t=t_1+1}^{T-1} \frac{\rho^{t-t_1} - 1}{\rho - 1} = \frac{\rho^{T-t_1} - \rho - (T - t_1 - 1)(\rho - 1)}{(\rho - 1)^2}$$

$$\Longleftarrow 2\rho^{T-t_1-2} \geq \frac{\rho^{T-t_1}}{(\rho - 1)^2} \iff \rho^2 - 4\rho - 2 \geq 0 \iff \rho \geq 2 + \sqrt{6}.$$

By our choice of $\rho$, we know condition (76) holds, and this completes our proof.

$\square$

## C    Extensions of Numerical Experiments for Different Problem Parameters

### C.1    Numerical Experiments on Spectral Norm of $\bar{A}$

In this section, we use the same experimental setup as in Section 8 for Lipschitz continuous basis functions. We examine the relationship between sample complexity and the spectral norm $\rho$. Specifically, we set $T = 100$, $p = 0.75$, and $n = 3$. To eliminate randomness in the spectral norm $\|\bar{A}\|_2$, we assign the singular values of $\bar{A}$ as $\sigma_1 = \cdots = \sigma_n = \rho$, where $\rho \in \{0.5, 0.95, 1.5\}$. In the case where $\rho = 1.5$, we terminate the simulation when $\|x_t\|_2 \geq 10^{14}$, as this indicates that the trajectory diverges to infinity, causing numerical issues for the CVX solver.

The results, presented in Figure 5, reveal that the required sample complexity increases slightly as $\rho$ increases from 0.5 to 0.95, which is consistent with Theorem 6. In addition, when $\rho = 1.5$, the system is not asymptotically stable, violating Assumption 5. The resulting divergence of the system state ($\|x_t\|_2 \to \infty$) leads to numerical instabilities in computing the estimator (3). However, it is possible that the estimator in (3) could still achieve exact recovery for large values of $\rho$ if a numerically stable method is employed for optimization. This does not contradict our theoretical findings, as Theorem 6 provides only a sufficient condition for exact recovery rather than a necessary one.

### C.2    Numerical Experiments with Small Probabilistic Sparsity Model

In this section, we repeat the experiments in Figure 3 with $p \in \{0.001, 0.1, 0.3\}$ and $n = 5$. The results are presented in Figure 6. We observe that the predictor fails to recover the ground truth within 500 steps when $p = 0.001$, whereas it successfully converges when $p = 0.1$ and $0.3$. Notably, both the loss gap and the optimality certificate are equal to zero in the case of $p = 0.001$. This occurs because multiple global

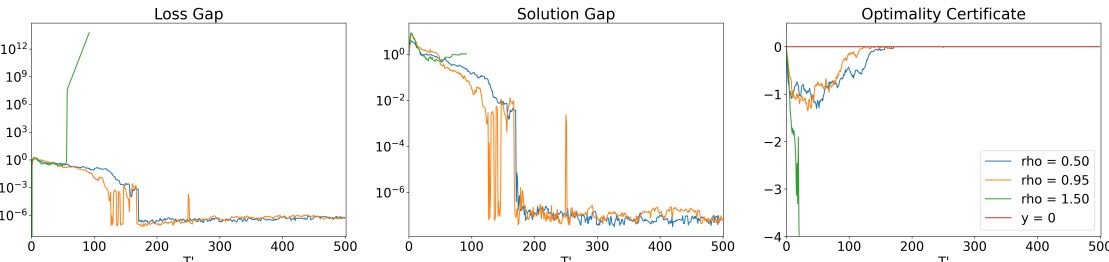

Figure 5: Loss gap, solution gap and optimality certificate of the Lipschitz basis function case with spectral norm $\rho = 0.5, 0.95$ and $1.5$.

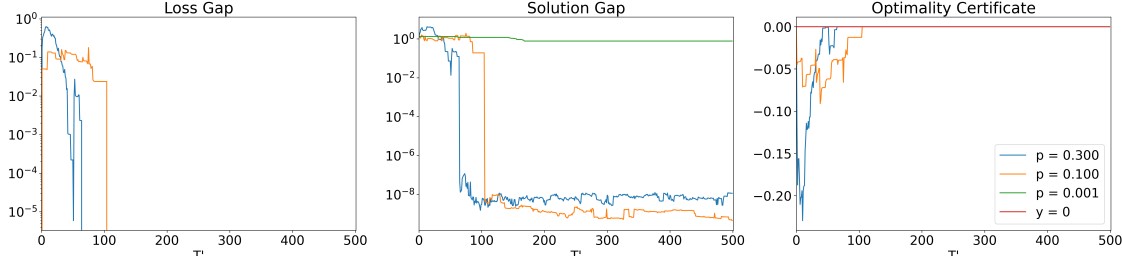

Figure 6: Loss gap, solution gap and optimality certificate of the Lipschitz basis function case with attack/disturbance probability $p = 0.001, 0.1$ and $0.3$. Note that the loss gap and the optimality certificate for the case when $p = 0.001$ is always equal to 0.

solutions exist, causing the estimator to fail in recovering the unique ground truth solution within 500 iterations. However, given a larger number of samples, the algorithm will eventually converge to the correct solution. That said, the primary focus of this paper is the regime where $p > 0.5$. When $p$ is very small or zero, learning the system falls within the classical control theory framework, where it is well established that an artificial excitation signal must be introduced to facilitate learning. The necessity of excitation signals in nearly deterministic systems is well documented in the control literature. For example, consider the linear system $x_{t+1} = Ax_t$, where the objective is to learn $A$ from observations of $x_t$. If the initial condition $x_0$ is zero, then $x_t$ remains zero for all $t$, making it impossible to infer $A$. To circumvent this issue, an artificial excitation signal is typically introduced, yielding a system of the form $x_{t+1} = Ax_t + w_t$ where $w_t$ is, for instance, Gaussian noise. Interestingly, when $p$ is sufficiently large, the adversarial attack itself serves as an effective excitation signal, aiding in the learning process by introducing necessary perturbations into the system.

## C.3 Numerical Experiments with Sparse $\bar{A}$

In this section, we replicate the experiments presented in Figure 3, using a sparse ground truth matrix $\bar{A}$. Specifically, we generate a sparse matrix $\bar{A}$ as a tridiagonal matrix, where each entry $\bar{A}_{i,j}$ is set to zero whenever $|i - j| > 1$. We conduct the experiments for Lipschitz basis functions with nonzero disturbance probabilities $p \in 0.7, 0.8, 0.85$ and system dimension $n = 10$. Additionally, we extend the simulation period to $T = 1000$, compared to $T = 500$ in the previous experiments. To improve computational efficiency, we solve the optimization problem (3) every 10time steps. Consequently, the plots exhibit discrete jumps corresponding to time periods that are multiples of ten. The loss gap is omitted from the figures since the estimator is computed only at a subset of time points. Figure 7 suggests that exact recovery is achieved despite the sparse structure of the ground truth matrix $\bar{A}$. This result is expected, as our theoretical analysis does not impose any dependency on the sparsity structure of $\bar{A}$. Beyond demonstrating robustness, the non-smooth objective function in (3) also acts as an implicit regularization mechanism that aligns with the specific matrix structure.

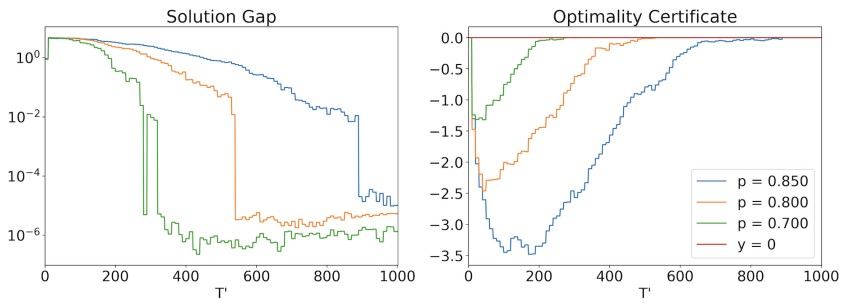

Figure 7: Solution gap and optimality certificate of the Lipschitz basis function case with $p \in \{0.7, 0.8, 0.85\}$ and $n = m = 10$.

### C.4 Numerical Experiments with Larger Order Systems

In this section, we extend the experiments presented in Figure 4 to significantly higher-order dynamical systems and a larger number of basis functions. Specifically, we consider system dimensions and basis function counts of $(n, m) \in (10, 20), (25, 50), (50, 100)$. The probability of a nonzero disturbance occurring is set to $p = 0.6$. Additionally, we increase the simulation period to $T = 1100$, compared to $T = 500$ in the previous experiments. To optimize computational efficiency, we solve the optimization problem (3) every 100 time periods. As a result, the plots exhibit discrete jumps corresponding to time periods that are multiples of 100. Due to this sampling strategy, the loss gap is omitted from the figures, as the estimator is computed only at a subset of time points.

In Figure 8, we observe that exact recovery is achieved even for large-scale system identification problems, where both the system dimension and the number of basis functions are significantly high. However, achieving exact recovery requires the system to evolve over a sufficiently long time horizon, as indicated by our theoretical results.

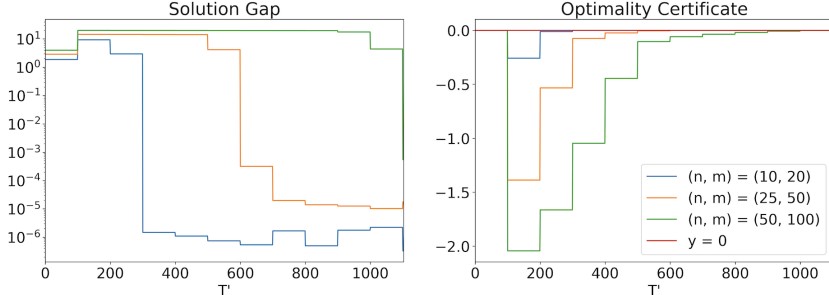

Figure 8: Loss gap, solution gap, and optimality certificate of the Lipschitz basis function case with dimension $(n, m) = (10, 20), (25, 50)$ and $(50, 100)$.

