# OpenReview forum: "Exact Recovery Guarantees for Parameterized Nonlinear System Identification Problem under Sparse Disturbances or Semi-Oblivious Attacks"
_TMLR — Accepted by TMLR_

### Review · Reviewer_T8Dq · 2025-03-23

**Summary Of Contributions:**

This paper designs a LASSO-type estimator for the nonlinear system identification problem. The proposed approach could recovery the ground truth exactly with a finite time.

**Audience:**

Yes

**Claims And Evidence:**

Yes

**Requested Changes:**

I have some questions and suggestions. I would appreciate it if the authors could handle them.
1. Theorems 4-6 are not self-contained since the definition of $p$ is not clearly stated.
2. Why can a LASSO-type estimator help you recover the estimator? Could the author give some intuition?
3. It is well known that LASSO estimator improves the estimator performance when the ground-truth parameter is sparse. In this paper, the sparsity of the disturbance is assummed. Any relationship between these two, i.e., the sparsity of parameter and sparsity of disturbance?
4. In adversarial training, the standard assumption is that the magnitude of the attacks are bounded. The assumptions in this paper, like the zero mean and gaussian distribution, are not convincing enough.
5. Is the lower bound in Theorem 5 applied to both bounded and Lipschitz base functions?  Is it possible to derive the bound in terms of the parameters of the $\kappa$?

**Strengths And Weaknesses:**

Strengths:

1. The paper is well-written and easy to follow.

2. The theoretical contributions are clearly stated.

3. The paper gives non-asymptotic finite time analysis for the nonlinear identification system under a general disturbance model.


Weakness:

1.The given bound is not tight.

2. The attack type investigated in this paper is kind of confusing.

3. The essense of the recovery ability of the proposed method has not been discussed.

---

> ### Author Response · Authors · 2025-05-26
>
> Weakness:
>
> 1. The given bound is not tight.
>
> Response: We would like to thank the reviewer for the comment. We agree with the reviewer that there exists a gap between the complexity bound of Theorem 4 and Theorem 5. Our work serves as the first step to analyzing the LASSO-type estimator (3) for temporally correlated attacks and establishing the first exact recovery guarantee for this type of estimators in the setting of system identification. It is likely that the complexity lower bound in Theorem 5 is loose. This is because the main purpose of Theorem 5 is to show that the required number of time steps grows inversely with $p(1-p)$ and grows proportionally to $m$. In this work we have shown that learning the system "in finite time" is possible which is the first result in the literature of its kind. Our bound proves the learning in finite time. Finding the smallest finite time is an interring problem that is left as future work.
>
>
> 2. The attack type investigated in this paper is kind of confusing.
>
> Response: We would like to thank the reviewer for the comment. Basically, our work focused on sparse and semi-oblivious attacks. Our assumption on the attack type is mainly motivated by the observation that attacks that do not satisfy the assumption may be easily detected and nullified by the system operators. The attacker should refrain from utilizing extreme attack strategies as the operator can analyze the statistical behavior of the inputs and identify deviations from natural disturbances, flagging them as attacks. Thus, the risk for an attacker employing extreme attacks is that they increase the likelihood of detection and nullification by a well-designed defense mechanism. Consequently, a strategic adversary should avoid highly conspicuous attack patterns and instead focus on attacks that have a lower probability of detection. For example, if the attacks have nonzero mean, this will create a consistent shift of the system states and may be detected by statistical hypothesis testing techniques. As another example, if the disturbance can contain arbitrarily large values that go beyond the physical limits (e.g., the readings from measurement devices cannot be larger than the limits of devices), they will be identified by the system operators as anomalies and removed from the reported data. See our discussions in Section 4 for more details.
>
> In addition, when the attack or disturbance vectors are possibly dependent zero mean sub-Gaussian vectors with nonzero measure in every subspace in $\mathbb{R}^n$, they are the special case of the attack vectors considered in the paper. It can verified that zero mean i.i.d. sub-Gaussian vectors satisfy the Assumptions 1, 3 and 6 presented for bounded and Lipschitz basis functions.
>
>
> 3. The essence of the recovery ability of the proposed method has not been discussed.
>
> Response: The majority of the paper is on the recovery ability of the proposed estimator. We have shown that the estimator is able to recover the system in finite time in various settings and studied its complexity. Unfortunately, we were not able to understand what you mean by discussing the "essence" of the recovery ability. There are already several discussions in the paper about the estimator and therefore we are not sure what additional discussions would like us to add to the paper.
>
>
> Requested Changes:
>
> 1. Theorems 4-6 are not self-contained since the definition of $p$ is not clearly stated.
>
> Response: We would like to thank the reviewer for the comment. We have included the definition of $p$ in Theorems 4-6. Suppose that the disturbance vector $\bar{d}_t$ is nonzero with probability $p\in(0,1)$.
>
>
> 2. Why can a LASSO-type estimator help you recover the estimator? Could the author give some intuition?
>
> Response: We would like to thank the reviewer for the suggestion. The intuition behind the advantage of the LASSO-type estimator (3) is that these estimators are robust to 'sparse' attacks. In our work, the sparsity of attacks is defined through probability $p$ (Definition 1). Suppose that the attack vectors $\bar{d_{t}}$ are nonzero and large for a small number of time indices $t$. The least-squares estimator (2) will generally not be able to exactly recover the underlying parameter $A$ since the loss function is smooth with respect to the error term $x_{t+1} - Af(x_t)$. In contrast, we proved that the LASSO-type estimator is still able to exactly recover parameter $A$ given a sufficiently large number of time steps. Intuitively, this is guaranteed through the nonsmooth loss function in (3), which prefers solutions that result in a small number of nonzero errors $x_{t+1}-Af(x_t)$. This property of the LASSO-type estimator (3) fits better to our setting than the least-squares estimator (2). We have included the intuition in the revised manuscript.

---

> ### Author Response · Authors · 2025-05-26
>
> 3. It is well known that LASSO estimator improves the estimator performance when the ground-truth parameter is sparse. In this paper, the sparsity of the disturbance is assumed. Any relationship between these two, i.e., the sparsity of parameter and sparsity of disturbance?
>
> Response: We would like to thank the reviewer for the insightful comment. The nonsmooth loss function in (3) is able to guarantee the sparsity of error terms. In our formulation, the error term is defined as the disturbance $x_{t+1} - Af(x_t)$, which leads to solutions $A$ such that the disturbance is sparse. On the other hand, if the sparsity of the parameter $A$ is assumed, we could choose the error term to be $\sum_{i,j}|A_{ij}|$. This loss function will generate sparse solutions $A$ with a small number of nonzero entries. We can aslo add this term as a regularizer to the objective function if both terms are sparse.
>
>
> 4. In adversarial training, the standard assumption is that the magnitude of the attacks are bounded. The assumptions in this paper, like the zero mean and Gaussian distribution, are not convincing enough.
>
> Response: We would like to thank the reviewer for the comment. On the one hand, the case when the attacks have a bounded magnitude is a special case of the sub-Gaussian case in Assumption 6. To be more specific, suppose that $||\bar{d}_t||_2 \leq B$ for all $t$. In this case, we have $\ell_t \leq B$ for all $t$ and thus, the magnitude $\ell_t$ is sub-Gaussian with parameter $\sigma = B$. On the other hand, if the injected disturbances exhibit a nonzero mean, the system operator can easily detect the attacks by hypothesis testing techniques and the attacked system will be flagged as suspicious. Therefore, by maintaining a zero mean, semi-oblivious evade detection while still destabilizing the systems. We have included the above more detailed explanation in the revised manuscript.
>
>
> 5. Is the lower bound in Theorem 5 applied to both bounded and Lipschitz base functions? Is it possible to derive the bound in terms of the parameters of the $\kappa$?
>
> Response: We would like to thank the reviewer for the helpful comment. The result of Theorem 5 holds for both bounded and Lipschitz base functions. We have added this remark in the revised paper. The main purpose of Theorem 5 is to show that the required number of time steps grows inversely with $p(1-p)$, which is counter-intuitive since the small $p$ case is usually considered 'easy' case for the exact recovery. We leave the derivation of lower bounds that depend on $\kappa$ to future works.

---

### Review · Reviewer_8ZXG · 2025-04-04

**Summary Of Contributions:**

This paper focuses on identifying nonlinear systems, where the dynamics is parametered as linear combinations of some known basis functions and the state observations are corrupted by sparse-but-large disturbances. The non-smooth $l_2$ loss estimator is studied, and a ncecssary and sufficient trajecotry-dependent condition for exact recovery is provided. The authors then go on to provide sample complexity results for exact parameter recovery using this estimator under various conditions. In the first case, the nonlinear basis functions are assumed bounded and the noises are zero-mean. In the second case, the nonlinear basis functions are Lipschitz, the noises are sub-gaussian and the system is assumed stable. Utilizing more problem structure in the second case, a better sample complexity is achieved.

**Audience:**

Yes

**Claims And Evidence:**

Yes

**Requested Changes:**

It is critical to fix the main weakness mentioned. For other minor changes, please refer to the "Minor weaknesses" section.

**Strengths And Weaknesses:**

### Strength
In this paper, an all-round theoretical analysis is provided. The authors first provide a necessary and sufficient condition (Theorem 1) for exact parameter recovery. Moreover, a sample complexity lower bound (Theorem 5) is provided and two sample complexity upper bounds in two scenarios are provided. The theoretical assumptions and results are discussed in details. The effectiveness of the algorithm is verified by simulations.

### One main weakness
One major weakness of the lies in the flow of Section 4. Theorem 3 shows that when the system is not fully excited. However, the authors later introduced an assumption (Assumption 1) that the noises are zero-mean, which doesn't have anything to do with the intuition Theorem 3 provided. The reviewer strongly recommend the authors reorganize this secion. The reviewer can even consider removing Theorem 3, as it is well-known that learning the system is not always possible when the system is not fully excited.

### Minor weaknesses
- At the end of page 2, the paper writes that "A crucial first step in achieving this goal is accurately learning the system dynamics". I suggest revise the sentence since there exist some model-free control algorithms.
- In Assumption 5, should $L>1$?
- In proof of Theorem 1, the reviewer strongly recommend the authors to derive Equation (16) or cite related literature that contains the derivation.

---

> ### Author Response · Authors · 2025-05-26
>
> One main weakness:
>
> One major weakness of the lies in the flow of Section 4. Theorem 3 shows that when the system is not fully excited. However, the authors later introduced an assumption (Assumption 1) that the noises are zero-mean, which doesn't have anything to do with the intuition Theorem 3 provided. The reviewer strongly recommend the authors reorganize this section. The reviewer can even consider removing Theorem 3, as it is well-known that learning the system is not always possible when the system is not fully excited.
>
>
> Response: We would like to thank the reviewer for the constructive comment. We agree with the review that Theorem 3 does not provide a direct intuition for Assumption 1. Indeed, the theorem provides the motivation for the non-degenerate condition (Assumption 3) in Section 5. Under Assumption 3, it is guaranteed that the disturbance $d_t$ will span the full space when the number of data points is large enough. We will follow the reviewer's suggestion and revise the presentation of Sections 4-5.
>
>
> Minor weaknesses:
>
> 1. At the end of page 2, the paper writes that "A crucial first step in achieving this goal is accurately learning the system dynamics". I suggest revise the sentence since there exist some model-free control algorithms.
>
> Response: We would like to thank the reviewer for the suggestion. We have revised the sentence to: "Under safety-critical constraints, it is generally necessary to first learn the system dynamics before applying a control strategy, since applying an inadequate policy could shift the system states beyond safe limits, potentially leading to instability; see Section 2 and Moerland et al. (2023) for more detailed discussions. Therefore, to avoid the potential risks, a crucial first step in achieving the goal of optimal control under adversarial attacks is accurately learning the system dynamics."
>
>
> 2. In Assumption 5, should $L > 1$?
>
> Response: We would like to thank the reviewer for the insightful comment. It is not necessary to have $L>1$ in Assumption 5. Instead, the assumption requires $\rho L < 1$, which will serve as a stability assumption for the system (1). To provide an intuition to this assumption, suppose that $c>0$ is a constant. We can substitute $A$ and $f$ with $cA$ and $f / c$, respectively. This substitution will not change system (1). The constants $\rho$ and $L$ will become $c\rho$ and $L / c$, respectively; while their product $\rho L$ remains unchanged. This is the intuition why we only require the assumption $\rho < 1 / L$ instead separate assumptions on $\rho$ and $L$.
>
>
> 3. In proof of Theorem 1, the reviewer strongly recommend the authors to derive Equation (16) or cite related literature that contains the derivation.
>
> Response: We appreciate the reviewer's suggestion. We have include the reference to Theorem 4.5 in the following preprint:
>
> Clason, Christian. "Nonsmooth analysis and optimization." arXiv preprint arXiv:1708.04180 (2017).

---

### Review · Reviewer_d1h4 · 2025-05-13

**Summary Of Contributions:**

The paper studies the system identification problem where the system dynamics are considered to be non-linear, and the sequence of disturbances across different time-steps is sparse (i.e., non-zero at only a few time-steps). The non-linearity is modelled via a collection of $m$ known basis functions, hence estimating the system boils down to learning the coefficient matrix, denoted by $\bar{A}$. The focus is on exact recovery of $\bar{A}$ from an observed trajectory of length $T$. In this regard, a LASSO-type estimator of $\bar{A}$ is proposed which seeks to minimize a sum of $\ell_2$ norm error terms for robustly recovering $\bar{A}$. The following theoretical results are then shown.

1.	Firstly, necessary and sufficient conditions are established under which $\bar{A}$ is: a global solution (Theorem 1) and, a unique solution (Theorem 2) of the proposed optimization problem. These are established using standard first order optimality conditions for convex problems, using the generalized Farkas lemma. These results do not make any stochastic assumption on the disturbances.

2.	Secondly, conditions on $T$ are derived for exact recovery of $\bar{A}$, under a stochastic disturbance model where the disturbance at each time $t$ is non-zero with probability $p \in (0,1) $. For this model, it is shown that the proposed estimator recovers $\bar{A}$ exactly provided $T$ is suitably large (depending polynomially on the system parameters). This is shown in Theorem 4 for bounded basis functions, and Theorem 6 for Lipschitz basis functions for stable systems.

3.	Theorem 5 shows that if $T$ is smaller a threshold, then under the assumptions of Theorem 5, the proposed optimization problem does not have a unique solution. Theorem 7 shows that if the system is not stable, then under the other assumptions of Theorem 6, $\bar{A}$ is not a solution of the proposed optimization problem.

**Audience:**

Yes

**Broader Impact Concerns:**

I do not have any concerns on the ethical implications of this work.

**Claims And Evidence:**

Yes

**Requested Changes:**

I have some comments/queries which are outlined below.

1.  The proof of Corollary 1 is trivial and could just be omitted with a remark that Cauchy-Schwartz inequality is used.

2.  In the statement of Corollary 3, isn’t the requirement that (5) holds redundant since (9) implies (5)?

3.  I may be missing something but in the proof of Corollary 3 I do not understand the final sentence. If (9) holds then how does it imply the logical condition in (8)?

4.  [Section 4, page 7]: As mentioned earlier, the two paragraphs following Definition 1 can be made more precise. For e.g. when p = 1 and $f$ is identity, finite-time (approximate) recovery of $\bar{A}$ is possible via the OLS, and is also the subject of recent results. If $f$ was not the identity, but of the form as in the present work (defined using known basis functions), then I suppose OLS could still be shown to yield approximate recovery results (but this should be checked).

5. In Theorem 4, it’s interesting that no assumptions on stability are needed for $\bar{A}$. Can this be explained conceptually? I guess the setting of Theorem 4 is a bit stringent as it disallows the linear case where $f$ would be identity.

6. In Theorem 4 (and elsewhere), I think $T = \Theta()$ should be replaced with $T = \Omega()$ since the latter symbol is the appropriate one for lower bounds.

7. [Appendix B.3]: In the proof of Corollary 2, in the equation immediately following the sentence “Then condition (5) implies that…”, I do not understand why the first equality holds.

8. [Appendix B.6]: In (37) a lower bound on $|\mathcal{K}’|$ is written but don’t we need an upper bound here?

**Strengths And Weaknesses:**

-----------------
Strengths
-----------------

1.  The paper is written well overall. The problem is well-defined, and a good overview of the literature on system identification is provided. The presentation of the results is clean and is easy to follow for the most part.

2.  While there are many recent results on finite-time guarantees for system identification, most of them are for linear systems. The present work considers non-linear systems but is also focused on a particular setup where the disturbances are sparse, i.e. the number of non-zero disturbances are small. This enables them to derive exact recovery guarantees for $\bar{A}$ provided T is sufficiently large.  In contrast, existing results typically assume the disturbance (or “excitation”) to be zero-mean subgaussian vectors with unit-variance entries and analyse the ordinary least squares estimator. So, the results therein involve approximately recovering the system matrix, with error rates of the order $T^{-1/2}$. I found the results in this work interesting and conceptually different from most existing results, barring the work of Yalcin et al. (2023) which studied this problem for linear systems.

3.	The theoretical results themselves are quite thorough and appear to be correct in general. I do have some questions from the proofs which are outlined later below.

--------------------
Weaknesses
-------------------

1.	In terms of the exposition, I feel that the comparison with existing finite-time system identification results for LTI systems could be written more clearly and might be somewhat confusing for readers not familiar with the literature. In particular, the setting there considers the excitation term (referred to as “disturbance” in the present work) to be zero-mean subgaussians with unit variance entries. The unit variance is actually w.l.o.g for LTI systems due to a scaling argument, as noted in (Simchowitz et al. 2018).  So that stochastic model is more general than the one in the present work (when $f$ is identity) and one cannot obtain exact recovery results in that setup (but rather, only approximate recovery is possible). The sparse disturbance model is quite specific and allows for exact recovery results. This distinction between “finite-time exact recovery” and “finite-time approximate recovery” can be highlighted in a clear manner, especially in the introduction and also on page 7 (Section 4) where the notion of finite-time learning seems to implicitly consider only exact recovery of $\bar{A}$ (which I think is incorrect).

2.	There are some technical arguments in the proofs and at other places which I could not understand, I have listed them later below.

---

> ### Author Response · Authors · 2025-05-26
>
> Weaknesses:
>
> 1. In terms of the exposition, I feel that the comparison with existing finite-time system identification results for LTI systems could be written more clearly and might be somewhat confusing for readers not familiar with the literature. In particular, the setting there considers the excitation term (referred to as “disturbance” in the present work) to be zero-mean subgaussians with unit variance entries. The unit variance is actually w.l.o.g for LTI systems due to a scaling argument, as noted in (Simchowitz et al. 2018). So that stochastic model is more general than the one in the present work (when $f$ is identity) and one cannot obtain exact recovery results in that setup (but rather, only approximate recovery is possible). The sparse disturbance model is quite specific and allows for exact recovery results. This distinction between “finite-time exact recovery” and “finite-time approximate recovery” can be highlighted in a clear manner, especially in the introduction and also on page 7 (Section 4) where the notion of finite-time learning seems to implicitly consider only exact recovery of $\tilde{A}$ (which I think is incorrect).
>
> Response: We would like to thank the reviewer for the insightful comment. We agree with the reviewer that the finite-time approximate is also an important topic for the system identification problem. The exact recovery, as mentioned by the reviewer, is generally not possible without additional assumptions on the disturbance. The exact recovery guarantee is important especially for safety-critically constrained control systems. In this work, we focused on the sparse disturbance model (Definition 1) and derived exact recovery guarantees under the disturbance model. The theoretical results under the disturbance model is the first attempt to establish exact recovery guarantees for systems with time-series structures, and we expect that results can be derived under other similar disturbance models. We have included discussions on the approximation recovery in the revised manuscript.
>
>
> 2. There are some technical arguments in the proofs and at other places which I could not understand, I have listed them later below.
>
> Response: We would like to thank the reviewer for the helpful suggestions and comments. We have addressed all of the reviewer's questions in the following responses. Please see the corresponding responses under each of the reviewer's questions.
>
>
> Requested Changes:
>
> 1. The proof of Corollary 1 is trivial and could just be omitted with a remark that Cauchy-Schwartz inequality is used.
>
> Response: We have followed the reviewer's suggestion and omitted the proof of Corollary 1.
>
>
> 2. In the statement of Corollary 3, isn’t the requirement that (5) holds redundant since (9) implies (5)?
>
> Response: We would like to thank the reviewer for catching this. We have removed the requirement of condition (5) in Corollary 3.
>
>
> 3. I may be missing something but in the proof of Corollary 3 I do not understand the final sentence. If (9) holds then how does it imply the logical condition in (8)?
>
> Response: We appreciate the reviewer for pointing out the confusing sentence in the proof. Under the assumption in (9), there does not exist nonzero $Z$ satisfying the left-hand-side equality in (8). Therefore, the logical condition in (8) is not checked for all nonzero $Z$ and thus, the condition in Theorem 2 holds under the assumption (9).
>
>
> 4. [Section 4, page 7]: As mentioned earlier, the two paragraphs following Definition 1 can be made more precise. For e.g. when p = 1 and $f$ is identity, finite-time (approximate) recovery of $\bar{A}$ is possible via the OLS, and is also the subject of recent results. If $f$ was not the identity, but of the form as in the present work (defined using known basis functions), then I suppose OLS could still be shown to yield approximate recovery results (but this should be checked).
>
> Response: We would like to thank the reviewer for the helpful suggestion. We have included a discussion on the difference between LSE estimators and LASSO-type estimators after Definition 1. We have included the discussion in the response for the ease of review. In addition, the failure of LSE is presented on page 8 for any basis function f under the probabilistic sparsity model for the disturbance vectors. Simchowitz et al. (2018) established that the optimal asympyotic convergence rate for the LSE is $1/\sqrt{T}$ for the linear and parametrized non-linear dynamical systems.

---

> ### Author Response · Authors · 2025-05-26
>
> 5. In Theorem 4, it’s interesting that no assumptions on stability are needed for $\bar{A}$. Can this be explained conceptually? I guess the setting of Theorem 4 is a bit stringent as it disallows the linear case where $f$ would be identity.
>
> Response: We would like to thank the reviewer for the insightful comment. The reason that no stability assumption is required in this case is because the basis function is bounded and the status $f(x_t)$ are also bounded for all $t$. When the basis functions are bounded, the system will not explode within finite time. So, this is in line with bounded-input bounded-output stability rather than asymptotic stability. We utilized this condition in our statistical analysis and derived sample complexity upper bounds based on the bound $B$.
>
> We agree with the reviewer that the bounded basis setting will exclude the linear basis function. In this work, the linear basis function case has been later studied under the Lipschitz basis function setting in Section 6 (the linear case satisfies the Lipschitz condition). The bounded basis function case is mainly motivated by applications in electricity networks, where the basis functions are triangular functions as a result of physical laws.
>
>
> 6. In Theorem 4 (and elsewhere), I think $T=\Theta()$ should be replaced with $T=\Omega()$ since the latter symbol is the appropriate one for lower bounds.
>
> Response: We would like to thank the reviewer for the suggestion. We would like to point out that the conditions in this work are in the form of $T \geq \Theta()$ instead of $T=\Theta()$. The former condition is indeed equivalent to the reviewer's proposed condition $T=\Omega()$.
>
>
> 7. [Appendix B.3]: In the proof of Corollary 2, in the equation immediately following the sentence “Then condition (5) implies that…”, I do not understand why the first equality holds.
>
> Response: For the ease of explanation, we consider the case when $\mathcal{K}=\{0\}$ is a singleton and we denote $d := \hat{d}_0$ and $f := f(x_0)$. In this case, we choose $Z=df^T / ||df^T||_F$. Additionally, we can calculate that $||df^T||_F^2 = ||d||_F^2 ||f||_F^2$. Substituting into the right-hand-side of the equality, we have the right-hand-side is equal to $d^T (df^T) / ||df^T||_F f = ||d||_F^2 ||f||_F^2 / ||df^T||_F = ||df^T||_F$. Therefore, the equality holds with our choice of $Z$.
>
>
> 8. [Appendix B.6]: In (37) a lower bound on $|\mathcal{K}'|$ is written but don’t we need an upper bound here?
>
> Response: We would like to thank the reviewer for pointing out the confusion in the proof. Indeed, the coefficient before $|\mathcal{K}'|$ ($\frac{\nu^2}{2} - \frac{\nu}{\sqrt{m}B}$) is negative due to condition (34). We have included the explanation in the revised proof.

---

> > ### Comment · Reviewer_d1h4 · 2025-05-29
> > **response to authors**
> >
> > Thanks for the clarifications, I am fine with all of them. Just a small remark/suggestion:
> >
> > Regarding the usage of asymptotic notation, I think its a bit strange to use inequality with these symbols, e.g, $T \geq \Theta(\cdot)$, since these are meant to be used via "=". So if you want to give an asymptotic lower bound on $T$, then $T = \Omega()$ would be appropriate, and also consistent with the definition of this symbol.

---

> > > ### Author Response · Authors · 2025-05-30
> > > **Response to the reviewer**
> > >
> > > We would like to thank the reviewer for the response! We have made changes according to the reviewer's suggection and updated the manuscript.

---

### Decision · Action_Editor_1pH3 · 2025-07-15

**Recommendation:** Accept as is

**Audience:**

Yes

**Audience Explanation:**

Yes, the paper will be of interest to researchers in TMLR who work on control theory, reinforcement learning and sparse recovery.

**Claims And Evidence:**

Yes

**Claims Explanation:**

All of the reviewers consider the findings of the paper to be sound and in line with the claims made in the abstract.